# Mixed-Sample SGD: an End-to-end Analysis of Supervised Transfer Learning

**Yuyang Deng**
Columbia University, Statistics
yd2824@columbia.edu

**Samory Kpotufe**
Columbia University, Statistics
skk2175@columbia.edu

https://github.com/matthewgo2009/mixed-sample-sgd

## Abstract

Theoretical works on supervised transfer learning (STL)—where the learner has access to labeled samples from both source and target distributions—have for the most part focused on statistical aspects of the problem, while efficient optimization has received less attention. We consider the problem of designing an SGD procedure for STL that alternates sampling between source and target data, while maintaining statistical transfer guarantees without prior knowledge of the quality of the source data. A main algorithmic difficulty is in understanding how to design such an adaptive sub-sampling mechanism at each SGD step, to automatically gain from the source when it is informative, or bias towards the target and avoid negative transfer when the source is less informative.

We show that, such a mixed-sample SGD procedure is feasible for general prediction tasks with convex losses, rooted in tracking an abstract sequence of constrained convex programs that serve to maintain the desired transfer guarantees. We instantiate these results in the concrete setting of linear regression with square loss, and show that the procedure converges, with $1/\sqrt{T}$ rate, to a solution whose statistical performance on the target is adaptive to the a priori unknown quality of the source. Experiments with synthetic and real datasets support the theory.

## 1 Introduction

In supervised transfer learning (STL), some amount of target data is to be complemented by a usually larger amount of related *source* data towards training a predictor. A characteristic problem to be solved is whether and how much to bias towards the source or target data without prior knowledge of the predictive quality of the source data for the target task. Many recent theoretical works on STL have yielded important insights into general approaches that may guarantee good target performance. Our main aim in this work is to understand the extent to which such insights may be incorporated into actual efficient procedures, in particular, practical *stochastic gradient descent* (SGD) type procedures, while maintaining good statistical guarantees for transfer.

For background, theoretical approaches for STL often take the form of penalized or constrained risk minimization—e.g., minimizing empirical risk on source subject to low target risk, or vice versa—-or more generally, some type of weighted risk minimization that aims to favor either source or target data, whichever is most beneficial (which is not usually known a priori). For example, let $P$ and $Q$ denote source and target distributions respectively, a typical approach, say in linear regression, would be to consider a weighted objective of the form[1] $\hat{R}_P(\theta) + \lambda \hat{R}_Q(\theta)$ and solve for choice $\lambda^*$ so that $\theta^* = \theta^*(\lambda^*)$ has small target risk $R_Q(\theta)$.

---

[1] Equivalently, of the form $\alpha \hat{R}_P(\theta) + (1-\alpha)\hat{R}_Q(\theta)$ for $\alpha \in [0,1]$, $\alpha = 1 - \lambda/(1+\lambda)$.

39th Conference on Neural Information Processing Systems (NeurIPS 2025).

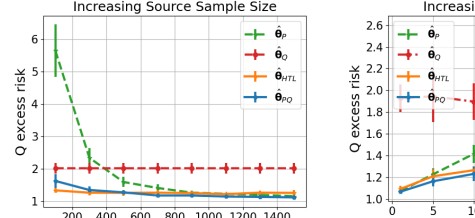
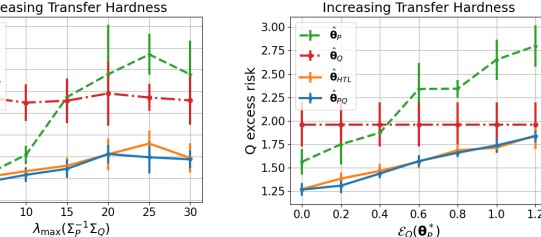

Figure 1: Simulation Results with Gaussian data, illustrating our guarantees that $\mathcal{E}(\hat{\theta}_{PQ}) < \min\{\mathcal{E}(\hat{\theta}_P), \mathcal{E}(\hat{\theta}_Q)\}$. $\hat{\theta}_{HTL}$ denotes the Hypothesis Transfer Learning (HTL). (Left) we fix $P, Q$ while increasing $n_P$, or (Middle) and (Right) we fix $n_P$, and push $P$ far from $Q$ as controlled by $\lambda_{\max}\left(\Sigma_P^{-1}\Sigma_Q\right)$ and $\mathcal{E}_Q(\theta_P^*)$. The source is least informative, i.e. source ERM $\hat{\theta}_P$ is worse than target ERM $\hat{\theta}_Q$, when either $n_P$ is too small (Left), or as $\lambda_{\max}\left(\Sigma_P^{-1}\Sigma_Q\right)$ or $\mathcal{E}_Q(\theta_P^*)$ is large (Middle and Right); we see that our method $\hat{\theta}_{PQ}$ automatically adapts to either situations and avoids negative transfer. HTL $\hat{\theta}_{HTL}$ yields results comparable to our method but it needs expensive cross-validation process to choose a proper bias parameter.

While many positive results have been derived over the years, they typically concern the target risk of the solution, upon a good choice of weights (i.e., $\lambda^*$), but do not address the computational aspects of the problem. For instance, choosing $\lambda$ (or any similar bias parameter) by cross-validation on the data (target and or source data) can be expensive as it involves many optimization passes over the combined data. Constrained risk minimization approaches, e.g., of the form $\min \hat{R}_P(\theta)$ s.t. $\hat{R}_Q(\theta) \lesssim \epsilon$ can similarly be expensive in maintaining the constraint (typically via expensive projection steps) through optimization iterations. This leaves open the extent to which such solutions may be achieved efficiently while at the same time maintaining strong statistical guarantees on target risk, adaptively to whether source or target datasets happen to be most informative for the target task.

We initiate the study of these questions in the context prediction tasks with convex losses, and propose a variant of SGD that alternates between sampling the source and target data at a sampling rate that changes according to a parameter $\lambda_t$ that automatically tracks the predictive quality of the source data. That is, for each SGD step $\theta_{t+1} = \theta_t - \eta\tilde{\nabla}\bar{R}(\theta_t; \lambda_t)$, the stochastic gradient estimates the gradient of an averaged empirical risk $\bar{R}(\theta; \lambda_t) \equiv \hat{R}_P(\theta) + \lambda_t\hat{R}_Q(\theta)$ depending on $\lambda_t$. Our main insight into the iterative choices of $\lambda_t$, evident in the analysis, is to let the stochastic gradient steps effectively *track a sequence of convex constrained objectives* (or CP for *convex program*) of the form

$$\min_\theta \hat{R}_P(\theta) \text{ s.t. } \hat{R}_Q(\theta) \leq \hat{R}_Q(\theta_{Q,t}) + \text{ slack}, \tag{1}$$

where $\theta_{Q,t} \xrightarrow{t\to\infty} \hat{\theta}_Q \doteq \arg\min_\theta \hat{R}_Q(\theta)$, i.e., $\theta_{Q,t}$ estimates the $Q$-ERM $\hat{\theta}_Q$ in parallel. The adaptive choice of sampling rate $\lambda_t$ is then chosen to track the sequence of max-min solutions of the corresponding Lagrangians $\mathcal{L}_t(\lambda, \theta) \approx \bar{R}(\theta, \lambda) - \lambda \cdot (\hat{R}_Q(\theta_{Q,t}) + \text{ slack})$.

On one hand, such a mixed-sample SGD solution replaces expensive cross-validation for the choice of bias parameter with the iterative choices of $\lambda_t$, and also avoids costly projections onto constraint sets. On the statistical side, we can show that the solution $\tilde{\theta}_{P,Q}$ of the limiting CP—i.e., replacing $\theta_{Q,t}$ in (1) with its limit $\hat{\theta}_Q$—achieve near optimal statistical guarantees for transfer whenever the setting, including loss functions, admit certain uniform concentration guarantees on empirical risk measures. Such statistical transfer guarantees are then shown to be inherited by the mixed-sample SGD solution $\hat{\theta}_{P,Q}$ which converges in risk to $\tilde{\theta}_{P,Q}$ at a typical rate of $O(1/\sqrt{t})$.

The main difficulty in the analysis is in showing convergence in $\hat{R}_P$ of $\hat{\theta}_{P,Q}$ to $\tilde{\theta}_{P,Q}$, while the statistical analysis of $\tilde{\theta}_{P,Q}$ combines insights from recent works on STL with either constrained or penalized objectives [1, 2, 3, 4, 5]. For intuition on technical difficulties, we note that recent classical works on SGD for CP's [6] rely heavily on the assumption that constraint sets are bounded, in order to at least approximately maintain constraints at each iteration via cheaper projections onto $\ell_2$ balls. We have to proceed differently as we consider general convex settings with potentially unbounded constraint sets (e.g., linear regression with non-invertible covariance in over-parametrized regimes). Our analysis instead relies on carefully tracking how far iterates $\theta_t$ may deviate from the constraint set, given the deviation of the first iterate and the internal variance of stochastic steps. Furthermore,

such control on the deviation of iterates is further complicated by the fact that, unlike in classical results such as [6], we are dealing with an evolving sequence of constraint sets given in terms of $\theta_{Q,t}$ which is being computed by a parallel SGD.

For the sake of presentation, we will focus on the concrete case of linear regression with square loss in the main paper, while the analysis for general losses, including surrogate losses for classification, is given in the appendix where we present the generalization guarantee for general convex losses in terms of Rademacher complexity. In the case of linear regression covered in the main text, the guarantees are immediately interpretable. Statistical guarantees take the form

$$R_Q(\hat{\theta}_{P,Q}) \lesssim R_Q(\theta_Q^*) + \min\{\epsilon_Q, \tilde{\epsilon}_P\},$$

where $\epsilon_Q, \tilde{\epsilon}_P$ are respectively, the best rate achievable by using the $Q$ data alone, and the best transfer rate achievable by using the $P$ data alone. In other words, the mixed-samples SGD solution is guaranteed to automatically bias towards whichever of the two samples is most predictive.

Many experimental results supporting these claims are presented in the main paper.

**Further Background.** The goal of reweighing source data relative to target data is rooted in early works on transfer learning and domain adaptation [see, e.g., 7, 8, 9, 10]. A main idea there is to find a weighting of the data that minimizes some notion of discrepancy between weighted source and target data. The actual target risk of the solution remains opaque in much of this line of work, as the theoretical analysis instead focuses on the well-posedness of the chosen notion of discrepancy and its estimability from data.

A different line of work, mostly focused on *covariate-shift settings* directly weighs source target data with estimated density ratios $dQ_X/dP_X$ and results in rates depending on the accuracy of such estimates in situations where the density ratio is well-defined [see,e.g., 11, 12].

More closely related, and often considering linear regression settings, the approach of *hypothesis transfer* aims to bias regression towards the solution from the source data via penalized objectives [see,e.g., 1, 3, 2, 13]. Recent works of [4, 5] consider constrained objectives for STL, mostly in classification settings. These various works are rather statistical in nature as they focus on understanding generalization properties of the solutions rather than their efficient estimation.

## 2 Setup and Preliminaries

**Data Distributions.** $P$ and $Q$ denote *source* and *target* distributions over $\mathcal{X} \times \mathcal{Y}$, $\mathcal{X} \subseteq \mathbb{R}^d$, $\mathcal{Y} \subseteq \mathbb{R}$.

**General Setting and Risks.** We consider a class of functions $f_\theta : \mathcal{X} \mapsto \mathcal{Y}$, parametrized by $\theta \in \Theta \subset \mathbb{R}^D$. For any distribution $\mu$, e.g., $P$ or $Q$, we let $R_\mu(\theta) \doteq \mathbb{E}_\mu \ell(f_\theta(X), Y)$, for a loss function $\ell$, and we let $\theta_\mu^*$ denote a risk minimizer. The *excess risk* is then defined as $\mathcal{E}_\mu(\theta) \doteq R_\mu(\theta) - R_\mu(\theta_\mu^*)$. The target excess risk $\mathcal{E}_Q(\cdot)$ is of main interest in STL.

### 2.1 Instantiation for Linear Settings.

As explained in the introduction, we focus on the case of linear regression with square loss in the main text. In this case we assume $\mathbb{E}_\mu[Y|X] = {\theta_\mu^*}^\top X$ for $\theta_\mu^*$ in $\mathbb{R}^d$.

**Assumption 1.** *For both distributions, we also assume that $Y - \mathbb{E}[Y|X]$ is $\sigma_y$-sub-gaussian and has zero mean, while $X$ is bounded, i.e., $\sup_{x \in \mathcal{X}} \|x\| < \infty$.*

**Relating $P$ to $Q$.** We use the notation $\Sigma_\mu \doteq \mathbb{E}_\mu X X^\top$ and $\|v\|_\Sigma \doteq v^\top \Sigma v$ for $v \in \mathbb{R}^d$, $\Sigma \in \mathbb{R}^{d \times d}$.

Recent results [14, 15, 16, 17, 18] have highlighted two essential quantities: (i) $\lambda_{\max}(\Sigma_P^{-1} \Sigma_Q)$, which characterizes the change in marginals $P_X \to Q_X$, and (ii) $\mathcal{E}_Q(\theta_P^*) \doteq \|\theta_P^* - \theta_Q^*\|_{\Sigma_Q}^2$, the change in optimal predictors. The first quantity remains relevant even when $\theta_P^* = \theta_Q^*$ and upper bounds error ratios $\|\theta - \theta_P^*\|_{\Sigma_Q}^2 / \|\theta - \theta_P^*\|_{\Sigma_P}^2$.

**Assumption 2.** $\Sigma_P$ *is full rank, while $\Sigma_Q$ may not be.*

The above assumption on $\Sigma_P$ may be somewhat relaxed, but is relevant in the transfer setting since otherwise $P$ may yield no information on $Q$ (in particular, $\lambda_{\max}(\Sigma_P^{-1} \Sigma_Q)$ is ill-defined).

**Empirical Quantities.** Throughout we assume that the learner has access to $n_P$ labeled samples $S_P \sim P^{n_P}$, and $n_Q$ labeled samples $S_Q \sim Q^{n_Q}$. We use $X_P$ and $X_Q$ to denote the set of feature vectors from $S_P$ and $S_Q$ respectively. We will also let $\mathbf{X}_P \in \mathbb{R}^{n_P \times d}$ and $\mathbf{X}_Q \in \mathbb{R}^{n_Q \times d}$ denote the corresponding data matrices, and $\mathbf{y}_P \in \mathbb{R}^{n_P}$ and $\mathbf{y}_Q \in \mathbb{R}^{n_Q}$ denote the corresponding vectors of labels. We use $S_{PQ}$ to denote the union of $S_P$ and $S_Q$.

Next, for any measure $\mu$, we let $\hat{\Sigma}_\mu$ denote the empirical counterpart of $\Sigma_\mu$ defined over $X_\mu$. Similarly, we let $\hat{R}_\mu(\theta) \doteq \frac{1}{n_\mu} \sum_{(x_i, y_i) \in S_\mu} (\theta^\top x_i - y_i)^2$. The following empirical risk minimizers (ERM's) are important to the analysis even though they are never directly computed:

**Definition 1.** *We let $\hat{\theta}_\mu \in \arg\min_{\theta \in \mathbb{R}^d} \hat{R}_\mu(\theta)$ denote the* minimum norm *ERM.*

In particular, $\hat{\theta}_P$ and $\hat{\theta}_Q$ will serve as baselines, i.e., we aim to outperform their risks under $Q$.

**Additional Notation.** Given a symmetric matrix $\Sigma$, we use $\lambda_{\min}^+(\Sigma)$ to denote its smallest non-zero eigenvalue. We write $a \lesssim b$ to indicate that $a \leq C \cdot b$ for some universal constant $C$.

## 3 Procedure

**Key Convex Programs.** As explained in the introduction, we aim to derive an efficient procedure to approximately track the following CP's, which, as we will later show, achieves rates of transfer automatically adaptive to whether the source or target data is most beneficial.

$$\min_{\theta \in \mathbb{R}^d} \hat{R}_P(\theta) \quad \text{subject to}: \quad \hat{R}_Q(\theta) \leq \hat{R}_Q(\theta_{Q,t}) + 6\epsilon_Q, \tag{2}$$

for $\theta_{Q,t} \xrightarrow{t \to \infty} \hat{\theta}_Q$. Intuitively, the above CP's aim for an interpolator between $\hat{\theta}_P$ and $\hat{\theta}_Q$ that constrains $Q$-excess risk to be of order no more than $\epsilon_Q = O(d/n_Q)$. The solution of the limiting CP will therefore be important to our analysis, and is highlighted in the following definition.

Consider the Lagrangian problem

$$\max_{\lambda \geq 0} \min_{\theta \in \mathbb{R}^d} \hat{R}_P(\theta) + \lambda(\hat{R}_Q(\theta) - \hat{R}_Q(\hat{\theta}_Q) - 6\epsilon_Q). \tag{3}$$

The saddle-point of the Lagrangian will be of importance in our anlysis.

**Definition 2.** *We let $(\lambda^*, \tilde{\theta}_{PQ})$ denote the solution of* (3) *above, whereby, by strong duality, $\tilde{\theta}_{PQ}$ is the solution of the limiting CP in* (2).

**Mixed-Samples SGD.** Algorithm 1 aims to approximate $(\lambda^*, \tilde{\theta}_{PQ})$ iteratively. However, since $\hat{\theta}_Q$ is unknown at the start of the procedure, the exact Lagrangian in (3) is undefined. Instead, the procedure maintains estimates of $\theta_{Q,t}$ of $\hat{\theta}_Q$ in parallel, and optimizes a time-varying Lagrangian $\mathcal{L}_t(\lambda, \theta) = \hat{R}_P(\theta) + \lambda(\hat{R}_Q(\theta) - \hat{R}_Q(\theta_{Q,t}) - 6\epsilon_Q)$. Notice that the iterative updates of $\lambda_t$ are in terms of stochastic estimates of the constraint violations, and therefore tracks the $Q$-risk of iterates $\theta_t$. Iterates $\lambda_t$ can thus be used in turn to adjust the sampling rates (see setting of $\xi_t$), i.e., to bias towards sampling from $S_P$ or $S_Q$.

## 4 Main Results: Instantiation for Linear Regression

We use the notation $M_x = \sup_{x \in \mathcal{X}} \|x\|$, $\hat{M}_y = \max_{(x,y) \in S_{PQ}} |y|$ and $\kappa_Q \doteq \frac{\lambda_{\max}(\hat{\Sigma}_Q)}{\lambda_{\min}^+(\hat{\Sigma}_Q)}$ throughout this section and subsequent sections. We start with the following definitions.

**Definition 3** (Key Lipschitz Parameters)**.** *Let $\rho \doteq \|\theta_0 - \tilde{\theta}_{PQ}\|$. We then define*

$$\hat{G}_\theta = \sup \left\{ \|\nabla \ell(\theta; x, y)\| : \|\theta - \tilde{\theta}_{PQ}\|^2 \leq 2\rho^2, (x, y) \in S_{PQ} \right\},$$

$$\hat{G}_\lambda = \sup \left\{ \begin{array}{l} |\ell(\theta; x, y) - \ell(\theta'; x, y) - 6\epsilon_Q| : \\ \|\theta - \tilde{\theta}_{PQ}\|^2 \leq 2\rho^2, (x, y) \in S_{PQ}, \|\theta'\|^2 \leq \left( \dfrac{1 + \log(T + 2\kappa_Q)}{\lambda_{\min}^+(\hat{\Sigma}_Q)} M_x \hat{M}_y \right)^2 \end{array} \right\}.$$

**Algorithm 1:** Mixed-Sample SGD

---

**Input:** $\theta_0 = \theta_{Q,0} = 0$, $\lambda_0 = 0$, stepsize $\{\alpha_t\}_{t=0}^{T-1}$ and $\eta$, $\gamma$, $\epsilon_Q$.

**for** $t = 0, \ldots, T-1$ **do**

    Draw $\xi_t \sim \text{Bernoulli}(\frac{1}{1+\lambda_t})$

    **if** $\xi_t = 1$ **then**

        Sample $(x_t, y_t)$ uniformly from $S_P$

        $\theta_{t+1} = \theta_t - \eta(1+\lambda_t)\nabla\ell(\theta_t; x_t, y_t)$

    **end**

    **else**

        Sample $(x_t, y_t)$ uniformly from $S_Q$

        $\theta_{t+1} = \theta_t - \eta(1+\lambda_t)\nabla\ell(\theta_t; x_t, y_t)$

    **end**

    Sample $(x_t, y_t)$ uniformly from $S_Q$

    $\lambda_{t+1} = [(1 - \gamma\eta)\lambda_t + \eta(\ell(\theta_t; x_t, y_t) - \ell(\theta_{Q,t}; x_t, y_t) - 6\epsilon_Q)]_+$

    $\theta_{Q,t+1} = \theta_{Q,t} - \alpha_t \nabla\ell(\theta_{Q,t}; x_t, y_t)$

**end**

$\hat{\theta}_{PQ} = \ell_2$ projection of $\frac{1}{T}\sum_{t=0}^{T-1}\theta_t$ onto the constraint set $\left\{\theta : \hat{R}_Q(\theta) - \hat{R}_Q(\theta_{Q,T}) \le 6\epsilon_Q - \epsilon_0\right\}$.

**Output:** $\hat{\theta}_{PQ}$

---

Our main results for Algorithm 1 are provided below.

**Theorem 1.** *Suppose parameters in Algorithm 1 are set as $\eta = \frac{c_\eta}{\sqrt{T}}$, $\gamma = c_\gamma \cdot \eta$, and $\alpha_t = \frac{1}{\lambda_{\min}^+(\hat{\Sigma}_Q)} \cdot \frac{1}{t+2\kappa_Q}$, $\epsilon_0 = \frac{C_{PQ}}{T+2\kappa_Q}$ for some $c_\eta$, $c_\gamma$ and $C_{PQ}$. Then, with probability $1 - 5\tau$ over $S_P$, $S_Q$ and the randomness in the procedure, the following holds for $c_\eta$ sufficiently small as a function of $(\hat{G}_\theta, \hat{G}_\lambda, \lambda^*, \rho)$, $c_\gamma \ge \hat{G}_\theta^2$ and $C_{PQ}$ as a functionof $(\hat{M}_x, \hat{M}_y, \log T, \lambda_{\min}^+(\hat{\Sigma}_Q))$. The returned solution satisfies*

$$\mathcal{E}_Q(\hat{\theta}_{PQ}) \lesssim \min\left\{\epsilon_Q, \lambda_{\max}\left(\Sigma_P^{-1}\Sigma_Q\right) \cdot \epsilon_P + \mathcal{E}_Q(\theta_P^*)\right\},$$

*for $\epsilon_P = c_0 \frac{\sigma_y^2(d + \log(1/\tau))}{n_P}$, $\epsilon_Q = c_0 \frac{\sigma_y^2(d + \log(1/\tau))}{n_Q}$ for some unversal constant $c_0 > 0$, provided a number of iterations*

$$T \gtrsim \left(\hat{G}_\theta + \hat{G}_\lambda\sqrt{\log\frac{1}{\tau}}\right)^2 \cdot \left(\frac{\frac{\hat{G}_\theta^2}{c_\eta} + \hat{G}_\theta\hat{G}_\lambda\sqrt{\log\frac{1}{\tau}}}{\lambda_{\min}^+(\hat{\Sigma}_Q)\epsilon_Q\epsilon_P} + \frac{\lambda^*\hat{G}_\lambda\sqrt{\log\frac{1}{\tau}} + \frac{\rho^2}{c_\eta}}{\epsilon_P}\right)^2.$$

The theorem is derived from both Theorem 2 of Section 5.1 (on optimization rates) and Theorem 3 of Section 5.2 on statistical rates. The exact requirements on $c_\eta$ are given in Theorem 2.

For completeness, in Section 4.1 we provide sample-dependent upper-bounds on intervening quantities, namely $\rho, \lambda^*, \hat{G}_\theta, \hat{G}_\lambda$, in terms of less opaque quantities.

**Adaptivity.** As so far discussed, the bounds of Theorem 1 guarantee that the procedure achieves a target risk always adaptive to whether the source or target is most beneficial: notice that if we were to use either the target sample alone or the source sample alone, we would be respectively achieving rates of the form $\mathcal{E}(\hat{\theta}_Q) \lesssim \epsilon_Q$, and $\mathcal{E}(\hat{\theta}_P) \lesssim \lambda_{\max}\left(\Sigma_P^{-1}\Sigma_Q\right) \cdot \epsilon_P + \mathcal{E}_Q(\theta_P^*)$. In other words, the returned solution $\hat{\theta}_{PQ}$ achieves a rate that interpolates between the two. This is illustrated by the simulations results of Figure 1 (the exact setting is described in detail in Section 6.1).

### 4.1 Sample-dependent Choices of Parameters $\eta$ and $\gamma$.

In this section we provide sample-dependent upper-bounds on $\rho$, $\hat{G}_\theta$, $\hat{G}_\lambda$ and $\lambda^*$ which drive the choice of $\eta$ and $\gamma$ in Theorem 1.

**Lemma 1** ($\rho$). *Assume $\hat{\Sigma}_P$ is invertible. The following upper bound holds:*

$$\rho^2 = \|\theta_0 - \tilde{\theta}_{PQ}\|^2 \le \left(\frac{\hat{\mathcal{E}}_P(\hat{\theta}_Q)}{\lambda_{\min}(\hat{\Sigma}_P)} + \frac{M_x\hat{M}_y}{\lambda_{\min}(\hat{\Sigma}_P)}\right)^2 \tag{4}$$

---

**Algorithm 2:** Warm-up

---

**Input:** $\theta_{Q,0} = 0$, stepsize $\{\alpha_t\}_{t=0}^{N-1}$
**for** $t = 0, \ldots, N - 1$ **do**
    |   Sample $(x_t, y_t)$ uniformly from $S_Q$
    |   $\theta_{Q,t+1} = \theta_{Q,t} - \alpha_t \nabla \ell(\theta_{Q,t}; x_t, y_t)$
**end**
**Output:** $\theta_{Q,N}$

---

*Furthermore, let $\theta_{Q,N}$ denote the output of the warmup procedure Algorithm 2 with stepsize $\alpha_t = \frac{1}{\lambda_{\min}^+(\hat{\Sigma}_Q)} \cdot \frac{1}{t + 2\kappa_Q}$. Then we can further bound (4) by the following quantity*

$$\frac{1}{2} \left( \frac{\|\nabla \hat{R}_P(\theta_{Q,N})\|^2 + \left( \frac{\lambda_{\max}(\hat{\Sigma}_P)}{\lambda_{\min}^+(\hat{\Sigma}_Q)} \right)^2 \|\nabla \hat{R}_Q(\theta_{Q,N})\|^2}{\lambda_{\min}^2(\hat{\Sigma}_P)} \right)^2 + 2 \frac{M_x^2 \hat{M}_y^2}{\lambda_{\min}^2(\hat{\Sigma}_P)}. \tag{5}$$

**Remark 1.** *Lemma 1 provides a computable upper bound of $\rho$, which requires a few steps of SGD to estimate $\hat{\theta}_Q$. As the number of steps $N$ increases, the $\|\nabla \hat{R}_Q(\theta_{Q,N})\|^2$ term in (5) tends to 0, and the whole bound becomes a tighter approximation of (4).*

**Lemma 2** ($\hat{G}_\theta$). *The following statement holds for $\hat{G}_\theta$: $\hat{G}_\theta \leq 3M_x^2 \rho + M_x \hat{M}_y$.*

**Lemma 3** ($\hat{G}_\lambda$). *The following statement holds for $\hat{G}_\lambda$:*

$$\hat{G}_\lambda \leq 18 \left( M_x^2 \rho^2 + \hat{M}_y + \frac{M_x^4 \hat{M}_y^2 (1 + \log(T + 2\kappa_Q))^2}{(\lambda_{\min}^+(\hat{\Sigma}_Q))^2} \right) + 6\epsilon_Q. \tag{6}$$

**Lemma 4** ($\lambda^*$). *Let $\lambda^*$ be defined in Definition 2. The following statement holds for $\lambda^*$:*

$$\lambda^* \leq \frac{\lambda_{\max}(\hat{\Sigma}_P)}{\lambda_{\min}^+(\hat{\Sigma}_Q)} + \frac{\|\nabla \hat{R}_P(\hat{\theta}_Q)\|}{2\sqrt{\lambda_{\min}^+(\hat{\Sigma}_Q)\epsilon_Q}}.$$

**Remark 2.** *In Lemma 4, the second term depends on the norm of $\nabla \hat{R}_P(\hat{\theta}_Q)$ divided by $\sqrt{\epsilon_Q}$. If $\hat{\theta}_P$ is close to $\hat{\theta}_Q$, then the second term becomes small. As in Lemma 1, one can also use Algorithm 2 to find an approximation of $\hat{\theta}_Q$, which yields a computable upper bound of $\lambda^*$*

## 5 Analysis Overview

In this section, we will provide more detailed convergence bound and generalization bound, as well as an overview of the analysis of our algorithm. Theorem 2 provides the convergence result of Algorithm 1, with more detailed parameter setup than Theorem 1. Theorem 3 provides the generalization guarantee of our algorithm.

### 5.1 Convergence Analysis

The following Theorem provides the convergence rate of Algorithm 1.

**Theorem 2.** *Let $\rho = \|\theta_0 - \tilde{\theta}_{PQ}\|$, $\tau \leq 0.1$, and $\sigma_{PQ}, c_\eta$ be some positive real numbers such that*

$$\sigma_{PQ}^2 \leq 256 \left( 1 + (\lambda^*)^2 + \rho^2 \right) \hat{G}_\theta^2, \ c_\eta \leq \min \left\{ \frac{\rho}{2\sqrt{2}\hat{G}_\lambda}, \frac{\rho}{16\sqrt{6}\sigma_{PQ}\sqrt{\log 2/\tau}}, \frac{\rho}{C_{PQ}} \right\},$$

*where $C_{PQ} = (1 + 2\kappa_Q)M_x^2 \hat{M}_y^2 + \frac{6\sigma_Q^2 \log(2/\tau)}{\lambda_{\min}^+(\hat{\Sigma}_Q)}$, and $\sigma_Q^2 = \left( \frac{\log(T + 2\kappa_Q)}{\lambda_{\min}^+(\hat{\Sigma}_Q)} M_x \hat{M}_y \right)^2$. Suppose the parameters in Algorithm 1 are set as $\eta = \frac{c_\eta}{\sqrt{T}}$, $\gamma = \frac{\hat{G}_\theta^2 c_\eta}{\sqrt{T}}$, $\alpha_t = \frac{1}{\lambda_{\min}^+(\hat{\Sigma}_Q)} \cdot \frac{1}{t + 2\kappa_Q}$ and $\epsilon_0 = \frac{C_{PQ}}{T + 2\kappa_Q}$. Assume $T \geq \max\{ \frac{3C_{PQ}\log T}{\epsilon_Q}, ( \frac{C_{PQ} \cdot (\log(T + 2\kappa_Q))^2}{\hat{G}_\lambda \sqrt{\lambda_{\min}^+(\hat{\Sigma}_Q)\epsilon_Q \log 1/\tau}} )^2 \}$, $\epsilon_Q \cdot \lambda_{\min}^+(\hat{\Sigma}_Q) \leq 1$ and $M_x \geq 1$. With*

*probability at least $1 - \tau$ over the randomness of Algorithm 1, the empirical risk of the returned solution satisfies:*

$$\hat{R}_P(\hat{\theta}_{PQ}) - \hat{R}_P(\tilde{\theta}_{PQ}) \lesssim \left( \hat{G}_\theta + \hat{G}_\lambda \sqrt{\log \frac{1}{\tau}} \right) \cdot \left( \frac{\frac{\hat{G}_\theta^2}{c_\eta} + \hat{G}_\theta \hat{G}_\lambda \sqrt{\log \frac{1}{\tau}}}{\lambda_{\min}^+(\hat{\Sigma}_Q)\epsilon_Q \sqrt{T}} + \frac{\lambda^* \hat{G}_\lambda \sqrt{\log \frac{1}{\tau}} + \frac{\rho^2}{c_\eta}}{\sqrt{T}} \right).$$

*Moreover, the total computational complexity of the algorithm is $O(dT)$ + time for projection.*

**Proof Sketch:** We define $\bar{\theta}_T \doteq \frac{1}{T} \sum_{t=0}^{T-1} \theta_t$ and $g(\theta) \doteq \hat{R}_Q(\theta) - \hat{R}_Q(\hat{\theta}_Q) - 6\epsilon_Q$. First, we inductively show that, if choose a properly small $\eta$, we can control all the iterates $(\theta_t, \lambda_t)$ as well as the final solution $\hat{\theta}_{PQ}$ to be stay around $\tilde{\theta}_{PQ}$ and $\lambda^*$. Hence, we can apply the $\hat{G}_\theta$ Lipschitzness on those iterates and decompose the risk as: $\hat{R}_P(\hat{\theta}_{PQ}) - \hat{R}_P(\tilde{\theta}_{PQ}) \leq \hat{G}_\theta \|\bar{\theta}_T - \hat{\theta}_{PQ}\| + \hat{R}_P(\bar{\theta}_T) - \hat{R}_P(\tilde{\theta}_{PQ})$. To further bound $\|\bar{\theta}_T - \hat{\theta}_{PQ}\|$ and $\hat{R}_P(\bar{\theta}_T) - \hat{R}_P(\tilde{\theta}_{PQ})$, we analyze the convergence of the Lagrangian function and obtain that

$$\hat{R}_P(\bar{\theta}_T) - \hat{R}_P(\tilde{\theta}_{PQ}) + \frac{(g(\bar{\theta}_T))^2}{2(\gamma + \frac{1}{\eta T})} \leq \frac{c_1}{\sqrt{T}} + \frac{c_2 \log T}{T} + c_3 \cdot |g(\bar{\theta}_T)| \tag{7}$$

for some real numbers $c_1, c_2, c_3$ depending on $T, c_\eta, \hat{G}_\theta, \hat{G}_\lambda, \rho, \lambda^*, \epsilon_Q$ and $\lambda_{\min}^+(\hat{\Sigma}_Q)$. Next, we will derive the lower and upper bound of $g(\bar{\theta}_T)$ in terms of $\|\bar{\theta}_T - \hat{\theta}_{PQ}\|$:

$$(g(\bar{\theta}_T))^2 \geq \frac{3}{2}\lambda_{\min}^+(\hat{\Sigma}_Q)\epsilon_Q \|\bar{\theta}_T - \hat{\theta}_{PQ}\|^2 - \Delta^2, g(\bar{\theta}_T) \leq \hat{G}_\theta \|\bar{\theta}_T - \hat{\theta}_{PQ}\|,$$

for $\Delta \doteq \epsilon_0 - (\hat{R}_Q(\theta_{Q,T}) - \hat{R}_Q(\hat{\theta}_Q))$ which captures the error from projecting $\bar{\theta}_T$ onto the *inexact* constraint set $\hat{R}_Q(\theta) - \hat{R}_Q(\theta_{Q,T}) - 6\epsilon_Q + \epsilon_0 \leq 0$. Plugging the above bounds together with $\hat{R}_P(\bar{\theta}_T) - \hat{R}_P(\tilde{\theta}_{PQ}) \geq -\hat{G}_\theta \|\bar{\theta}_T - \tilde{\theta}_{PQ}\|$ back to (7) one can solve an upper bound of $\|\bar{\theta}_T - \hat{\theta}_{PQ}\|$. Notice that (7) immediately gives an upper bound of $\hat{R}_P(\bar{\theta}_T) - \hat{R}_P(\tilde{\theta}_{PQ})$ which concludes the proof.

## 5.2 Generalization Analysis

The following result establishes the generalization guarantee of our algorithm.

**Theorem 3.** *Suppose $\hat{\theta}_{PQ}$ satisfies $\hat{R}_P(\hat{\theta}_{PQ}) - \hat{R}_P(\tilde{\theta}_{PQ}) \leq \epsilon_P$. Then with probability at least $1 - 4\tau$ over $S_P$ and $S_Q$, the excess risk of $\hat{\theta}_{PQ}$ on $Q$ satisfies:*

$$\mathcal{E}_Q(\hat{\theta}_{PQ}) \leq 26 \min \left\{ \epsilon_Q, \lambda_{\max} \left( \Sigma_P^{-1} \Sigma_Q \right) \epsilon_P + \mathcal{E}_Q(\theta_P^*) \right\}.$$

To prove Theorem 3, we first introduce some technical lemmas. The following lemma gives two information: (i) any $\theta$ in the constraint set of (2) will have a small $Q$ risk, and (ii) any $\theta$ that has a small $Q$ risk is covered by our constraint set with high probability.

**Lemma 5.** *With probability at least $1-2\tau$, the following holds: for any $\theta \in \{\theta : \|\theta - \hat{\theta}_Q\|_{\hat{\Sigma}_Q}^2 \leq 6\epsilon_Q\}$, we have $\mathcal{E}_Q(\theta) \leq 26\epsilon_Q$. For any $\theta$ such that $\mathcal{E}_Q(\theta) \leq \epsilon_Q$, it is in $\{\theta : \|\theta - \hat{\theta}_Q\|_{\hat{\Sigma}_Q}^2 \leq 6\epsilon_Q\}$.*

The next lemma shows that, if $\theta_P^*$ is in the constraint set of (2), any $\theta$ in the constraint set with a small empirical risk on $P$, then it should also have a small population risk on $P$.

**Lemma 6.** *If $\theta_P^* \in \{\theta : \|\theta - \hat{\theta}_Q\|_{\hat{\Sigma}_Q}^2 \leq 6\epsilon_Q\}$, then for any $\theta \in \{\theta : \|\theta - \hat{\theta}_Q\|_{\hat{\Sigma}_Q}^2 \leq 6\epsilon_Q\}$, with probability at least $1 - 2\tau$ we have $\mathcal{E}_P(\theta) \leq 4(\hat{R}_P(\theta) - \hat{R}_P(\tilde{\theta}_{PQ})) + 16\epsilon_P$.*

The next Lemma considers the situation where $\theta_P^*$ is not in the constraint set of (2). Due to the inclusive property of our constraint set (Lemma 5), we can claim that any $\theta$ in the constraint set has a smaller $Q$ excess risk than $\theta_P^*$ up tp some multiplicative universal constant.

**Lemma 7.** *If $\theta_P^* \notin \{\theta : \|\theta - \hat{\theta}_Q\|_{\hat{\Sigma}_Q}^2 \leq 6\epsilon_Q\}$, then for any $\theta \in \{\theta : \|\theta - \hat{\theta}_Q\|_{\hat{\Sigma}_Q}^2 \leq 6\epsilon_Q\}$, with probability at least $1 - 2\tau$ we have $\mathcal{E}_Q(\theta) \leq 26\mathcal{E}_Q(\theta_P^*)$.*

**Proof of Theorem 3.** First, if $\theta_P^* \in \{\theta : \|\theta - \hat{\theta}_Q\|_{\hat{\Sigma}_Q}^2 \le 6\epsilon_Q\}$, then since $\hat{R}_P(\hat{\theta}_{PQ}) - \hat{R}_P(\tilde{\theta}_{PQ}) \le \epsilon_P$, from Lemma 6 we know with probability at least $1 - 2\tau$, $\mathcal{E}_P(\hat{\theta}_{PQ}) \le 4\epsilon_P + 16\epsilon_P = 20\epsilon_P$. Hence $\mathcal{E}_Q(\hat{\theta}_{PQ}) = \|\hat{\theta}_{PQ} - \theta_Q^*\|_{\Sigma_Q}^2 \le \lambda_{\max}\left(\Sigma_P^{-1}\Sigma_Q\right) \cdot 20\epsilon_P + \mathcal{E}_Q(\theta_P^*)$. If $\theta_P^* \notin \{\theta : \|\theta - \hat{\theta}_Q\|_{\hat{\Sigma}_Q}^2 \le 6\epsilon_Q\}$, from Lemma 7 we know $\mathcal{E}_Q(\hat{\theta}_{PQ}) \le 26\mathcal{E}_Q(\theta_P^*)$. Hence we know with probability at least $1 - 2\tau$,

$$\mathcal{E}_Q(\hat{\theta}_{PQ}) \le 26\left(\lambda_{\max}\left(\Sigma_P^{-1}\Sigma_Q\right) \cdot \epsilon_P + \mathcal{E}_Q(\theta_P^*)\right).$$

On the other hand, since $\hat{\theta}_{PQ} \in \{\theta : \|\theta - \hat{\theta}_Q\|_{\hat{\Sigma}_Q}^2 \le 6\epsilon_Q\}$, from Lemma 5 we know $\mathcal{E}_Q(\hat{\theta}_{PQ}) \le 26\epsilon_Q$. Putting piece together we have with probability at least $1 - 4\tau$, it holds that

$$\mathcal{E}_Q(\hat{\theta}_{PQ}) \le 26\min\left\{\epsilon_Q, \lambda_{\max}\left(\Sigma_P^{-1}\Sigma_Q\right) \cdot \epsilon_P + \mathcal{E}_Q(\theta_P^*)\right\}.$$

$\square$

**Proof of Theorem 1.** In Theorem 2, choosing $T$ such that $\hat{R}_P(\hat{\theta}_{PQ}) - \hat{R}_P(\tilde{\theta}_{PQ}) \le \epsilon_P$, together with Theorem 3 will conclude the proof. $\square$

## 6 Experiments

In this section, we present the experimental results of our algorithm and the baseline algorithms on both synthetic and real-world datasets. We implement our algorithm using Python on an Intel i7-8700 CPU. We use the CVXPY [19] package to implement the projection step in Algorithm 1. The baseline algorithms we consider are source ERM, target ERM, projected SGD (PSGD) (widely used for solving constrained problems) and Hypothesis Transfer Learning (HTL). For HTL, we use 5-fold cross-validation on the target training data to select the best bias parameter. For PSGD, we also employ CVXPY to implement the projection steps. Throughout the figures in this section, we use $\hat{\theta}_{PSGD}$ and $\hat{\theta}_{HTL}$ to denote the model obtained by PSGD and HTL respectively. Due to page limit, we only present part of the results and defer additional ones to the appendix.

### 6.1 Regression Task on the Synthetic Dataset.

We start with the results on the synthetic dataset. Throughout this subsection, we set the model dimension $d$ to be 50. The distribution $P$ and $Q$ are set to be $d$-dimensional multivariate Gaussian with certain mean and covariance. The label is generated as $y = \theta_\mu^{*\top} x + \varepsilon$ for $\mu \in \{P, Q\}$, where $\varepsilon \sim \mathcal{N}(0, 1)$. We choose $\eta = 10^{-4}$, and iteration number $T = 2000 \cdot n_P$ for both our method and HTL. We use closed form OLS solution to compute source and target ERM model. The results are demonstrated in Figure 1 and 2. In Figure 1 (Left), we fix $n_Q = 100$ and vary $n_P$ from 100 to 1500. When $n_P$ is small, target ERM learning can work better, while as we increase $n_P$, the source data becomes more useful. Our method can adapt to these two regimes automatically. It can also be seen that the ERM procedure has large uncertainty (variance) when the training data is insufficient,

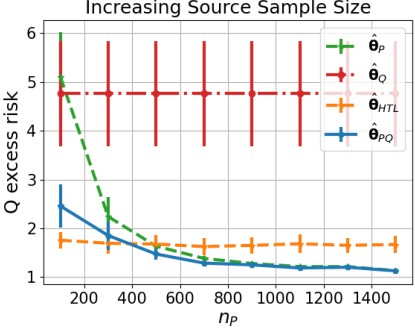
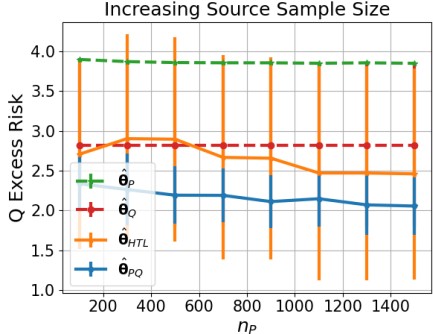

Figure 2: Linear regression results on the synthetic dataset with low-rank $\Sigma_Q$ or low-rank $\Sigma_Q$. (Left) When $\Sigma_Q$ is not full rank, the constraint set is then unbounded but our method still works well. (Right) When $\Sigma_P$ is not full rank, we set $w_Q^*$ far away from $P$'s range. In this case HTL still biases the model towards min-norm $\hat{w}_P$ and leads to a bad performance. However, our method is not influenced because it directly learns from $P$'s data.

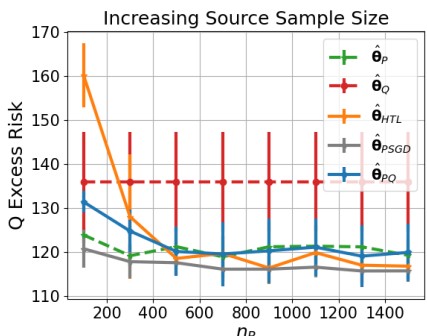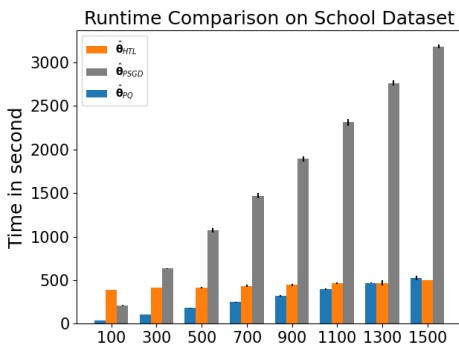

Figure 3: Linear regression results on the School dataset. (Left) Excess risk on $Q$. (Right) Runtime comparison. Mixed-Samples SGD $\hat{\theta}_{PQ}$ achieves $Q$-risk nearly the same of that of the ideal projection method $\hat{\theta}_{\mathrm{PSGD}}$, while achieving significantly faster runtime.

while our method is significantly more stable. In Figure 1 (Middle) and (Right), we fix $n_Q = 100$, $n_P = 500$, and gradually increase $\lambda_{\max}(\Sigma_P^{-1}\Sigma_Q)$ and $\mathcal{E}_Q(\theta_P^*)$, which measures the hardness of transfer. As the hardness increases, source ERM model performs poorly, while our method can always yield a model that is comparable to the better one between $\hat{\theta}_P$ and $\hat{\theta}_Q$. In Figure 2 (Left), we set $\Sigma_Q$ to be low rank and the target sample size to be extremely small ($n_Q = 50$). In this case, the constraint set is unbounded but our method still works well. The rate of our method is still adaptive, while due to the limited target data, HTL struggles to find a suitable regularization parameter through cross-validation, and hence loses the adaptivity. In Figure 2 (Right), We set $\Sigma_P$ to be low rank. We choose $n_Q = 10$ and set $w_Q^*$ far away from $P$'s range. We use SGD to compute $Q$ ERM model since matrix inversion is unstable in this case. HTL still computes the min-norm interpolator $\hat{w}_P$ as the reference model which has a large $Q$ risk. Due to the limited $Q$'s data, HTL often fails on finding proper bias parameter and still tries to bias the model towards $\hat{w}_P$ which leads to a bad performance. Even though in some cases it finds the correct bias parameter (close to zero), that means HTL fully abandons $\hat{w}_P$ and cannot gain from $P$'s data. However, our method is not influenced because it does not rely on the reference model to transfer knowledge, but directly learns on $P$'s data.

### 6.2  Regression Task on the School Dataset.

To demonstrate the performance of our method on real-world data, we conduct the experiments on the School Dataset [20]. The dataset contains student information from 139 schools. The input $x$ is a 27-dimensional vector containing student information and the label $y$ is the student's exam score. Following [21], we use the data points from the first 100 schools as the source domain and the rest as the target. We set $\eta = 10^{-4}$. We fix $n_Q = 100$ and vary $n_P$ from 100 to 1500. Figure 3 shows the MSE and runtime comparison with source ERM, target ERM, PSGD and HTL. We can see from Figure 3 (Left) that our method consistently outperforms source ERM, target ERM, and comparable to HTL and PSGD, and *automatically adapts* to the better rate between source and target ERM learning. PSGD yields performance comparable to ours since it solves the same objective we proposed in (2), but it requires significantly longer time, as shown in Figure 3 (Right).

### 6.3  Regression Task on the Berkeley Yearbook Dataset.

We conduct experiments on the Berkeley Yearbook Dataset [22]. The dataset contains the gray-scale portraits taken in different years. The input $x$ is the 512-dimensional vector feature extracted by ResNet18, and $y$ is the year the photo is taken (ranging from 1905 to 2013). We construct source and target tasks by varying the proportion of male and female photos. In the source dataset, 50% of the samples are drawn from male photos and 50% from female photos. For the target training and testing dataset, the ratio is adjusted to 75% male and 25% female. We choose $\eta = 10^{-4}$ for all algorithms, and iteration number $T = 2000 \cdot n_P$ for both our method and HTL, and $T = 500 \cdot n_P$ for PSGD since we found it can converge in fewer epochs. We use closed form OLS solution to compute source and target ERM model. We fix $n_Q = 100$ and vary $n_P$ from 500 to 1300. Due to the difficulty of the task and the limited target data, the target ERM model suffers from very large errors and is therefore omitted from Figure 4. We report the MSE comparison with source ERM and HTL, as well as a runtime comparison with HTL. We can see from the left sub-figure that our method

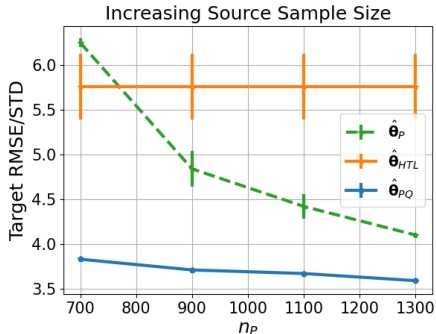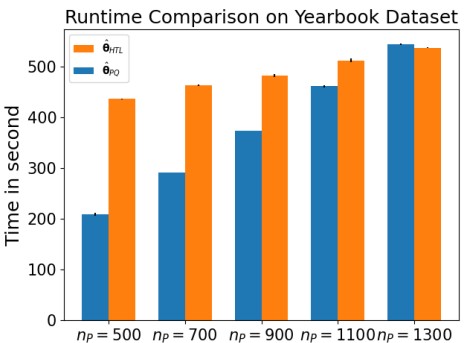

Figure 4: Linear regression on the Yearbook dataset. Input features of dimension 512 are extracted by ResNet18. (Left) Q excess risk. (Right) Runtime comparison. In the (Left) we omit the performance of target ERM since the target ERM fails on outputting a generalizable solution and results in the error over 4000 due to that the task is very difficult and target sample size is only 100. HTL ($\hat{\theta}_{\text{HTL}}$) has difficulty to adapt to the situation where source data are more helpful than target and results in large target error, while our algorithm ($\hat{\theta}_{PQ}$) gains from source data and significantly outperforms the competitors.

can consistently outperform source ERM, target ERM and HTL, and can *automatically adapt* to the better rate of source and target ERM learning. The right sub-figure shows that our algorithm achieves superior runtime performance compared to HTL when $n_P < 1300$, while when $n_P$ reaches 1300, HTL becomes the faster one. This difference arises from the inherent nature of the two algorithms: our method primarily optimizes over the source data, so the total number of iterations increases with $n_P$ grows, In contrast, HTL focuses on optimizing over the target data, using the source data only to compute a reference model. As a result, its runtime remains relatively stable as $n_P$ increases.

### 6.4 Binary Classification Task on the CIFAR-10 Dog vs Cat Dataset.

At last, to demonstrate that our algorithm can work for general convex losses, we conduct the binary classification experiment on the CIFAR-10 Dog vs Cat dataset [23], using linear classifier and logistic loss. We construct source and target tasks by varying the proportion of dog and cat images. In the source dataset, the ratio is $50\%$ dog and $50\%$ cat. For the target training and testing dataset, the ratio is adjusted to $80\%$ dog and $20\%$ cat. We use SGD with stepsize $10^{-5}$ and epoch number $1000$ to compute $P$ and $Q$ ERM model and HTL model. We choose stepsize for $\theta$ to be $10^{-5}$ and that for $\lambda$ to be $10^{-3}$, as well as $T = 2000 \cdot n_P$ in our method. We fix $n_Q = 50$ and vary $n_P$ from 100 to 1500. The results are reported in Figure 5. Target ERM

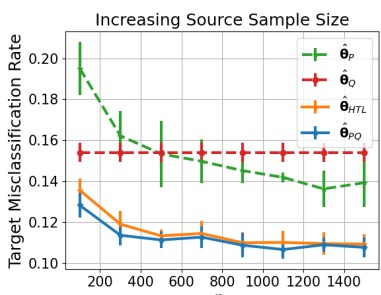

Figure 5: Binary classification results on the CIFAR Dog vs Cat dataset. We use linear classifier with logistic loss. Input features of dimension 512 are extracted by ResNet18. We fix $n_Q = 50$ and increase $n_P$. It verifies that our algorithm can work on the general loss.

model performs poorly since the target sample size is small, while our method can still enjoy an adaptive rate and slightly outperforms HTL.

## 7 Conclusion

In this paper, we propose a concrete optimization algorithm with provable convergence and generalization guarantee in supervised transfer learning. The analyzed procedure is a mixed-samples SGD approach that alternates between sampling from source or target data at an adaptive sampling rate. Both theoretical and experimental results show that our method is adaptive to whether the source or target data are most beneficial. This work aims to initiate the theoretical study of computationally efficient methods for transfer learning.

## 8 Acknowledgement

Both authors are grateful for support under NSF Award Number 2334997:CPS: Data Augmentation and Model Transfer for the Internet of Things.

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

**Algorithm 3:** Mixed-Sample SGD (Restated for General $\Theta$)

---

**Input:** $\theta_0 = \theta_{Q,0} = 0$, $\lambda_0 = 0$, stepsize $\{\alpha_t\}_{t=0}^{T-1}$ and $\eta, \gamma, \epsilon_Q$.
**for** $t = 0, \ldots, T-1$ **do**
    Draw $\xi_t \sim \text{Bernoulli}(\frac{1}{1+\lambda_t})$
    **if** $\xi_t = 1$ **then**
        Sample $(x_t, y_t)$ uniformly from $S_P$
        $\theta_{t+1} = \mathcal{P}_\Theta(\theta_t - \eta(1 + \lambda_t)\nabla\ell(\theta_t; x_t, y_t))$
    **end**
    **else**
        Sample $(x_t, y_t)$ uniformly from $S_Q$
        $\theta_{t+1} = \mathcal{P}_\Theta(\theta_t - \eta(1 + \lambda_t)\nabla\ell(\theta_t; x_t, y_t))$
    **end**
    Sample $(x_t, y_t)$ uniformly from $S_Q$
    $\lambda_{t+1} = [(1 - \gamma\eta)\lambda_t + \eta(\ell(\theta_t; x_t, y_t) - \ell(\theta_{Q,t}; x_t, y_t) - 3\epsilon_Q)]_+$
    $\theta_{Q,t+1} = \mathcal{P}_\Theta(\theta_{Q,t} - \alpha_t\nabla\ell(\theta_{Q,t}; x_t, y_t))$
**end**
$\hat{\theta}_{PQ} = \ell_2$ projection of $\frac{1}{T}\sum_{t=0}^{T-1}\theta_t$ onto the constraint set $\left\{\theta : \hat{R}_Q(\theta) - \hat{R}_Q(\theta_{Q,T}) \leq 3\epsilon_Q - \epsilon_0\right\}$.
**Output:** $\hat{\theta}_{PQ}$

---

# 9 Results for General Losses

In this section, we provide the results for general convex losses. We consider a hypothesis class $\mathcal{H} \doteq \{x \mapsto h_\theta(x) : \theta \in \Theta\}$. We make the following assumption on the loss function $\ell$.

**Assumption 3.** $\ell(\theta; x, y)$ *is $m_1$-strongly convex and $m_2$-smooth in $\theta$ for any $(x, y) \in S_{PQ}$.*

We define $\kappa \doteq m_1/m_2$ as the condition number of $\ell$.

We consider general $\Theta \subseteq \mathbb{R}^D$, possibly a strict subset, and therefore present a version of the algorithm that projects iterates back to $\Theta$, provided a projection operator $\mathcal{P}_\Theta$ (see Algorithm 3); when $\Theta = \mathbb{R}^D$, i.e. is unbounded, the operator reduces to identity mapping so Algorithm 3 reduces to Algorithm 1 in the main paper. Note that this projection operator is usually cheap for the common choices of $\Theta$: for example, for the sake of regularization in practice (e.g., ridge-type regularization), often $\Theta$ is an $\ell_2$ unit ball, whereby $\mathcal{P}_\Theta(\theta) = \theta/\max\{1, \|\theta\|\}$.

For the convergence analysis, we need the following definitions.

**Definition 4** (Key Lipschitz Parameters). *Let $\rho \doteq \left\|\theta_0 - \tilde{\theta}_{PQ}\right\|$. We then define*

$$
\hat{G}_\theta = \sup \left\{
\begin{array}{l}
\|\nabla\ell(\theta; x, y)\| : \left\|\theta - \tilde{\theta}_{PQ}\right\|^2 \leq 2\rho^2, (x, y) \in S_{PQ}, \\[2mm]
\|\theta\| \leq \dfrac{1 + \log(T + \kappa)}{m_1} \sup_{(x', y') \in S_Q} \|\nabla\ell(\theta_{Q,0}; x', y')\|.
\end{array}
\right\},
$$

$$
\hat{G}_\lambda = \sup \left\{
\begin{array}{l}
|\ell(\theta; x, y) - \ell(\theta'; x, y) - 3\epsilon_Q| : \left\|\theta - \tilde{\theta}_{PQ}\right\|^2 \leq 2\rho^2, (x, y) \in S_{PQ}, \\[2mm]
\|\theta'\| \leq \dfrac{1 + \log(T + \kappa)}{m_1} \sup_{(x', y') \in S_Q} \|\nabla\ell(\theta_{Q,0}; x', y')\|.
\end{array}
\right\}.
$$

The above definition captures the Lipschitzness of the risk function on the iterates generated during our algorithm proceeding.

**Definition 5** (See [6]). *For $\epsilon > 0$, let $r(\epsilon) \doteq \inf\left\{\|\nabla\hat{R}_Q(\theta)\|^2 : \theta \in \Theta, \hat{R}_Q(\theta) - \hat{R}_Q(\hat{\theta}_Q) = \epsilon\right\}$.*

This notion is used to control the distance between the return solution and solution before the last projection step, i.e., $\|\bar{\theta}_T - \hat{\theta}_{PQ}\|$ where $\bar{\theta}_T \doteq \frac{1}{T}\sum_{t=0}^{T-1}\theta_t$. In linear regression, we can explicitly compute it as $r(\epsilon) = \epsilon \cdot \lambda_{\min}^+(\hat{\Sigma}_Q)$.

**Theorem 4** (Strongly Convex and Smooth Result). *Let $\rho^2 = \|\theta_0 - \tilde{\theta}_{PQ}\|^2$, $\tau \leq 0.1$, and $\sigma_{PQ}^2, c_\eta$ be some positive numbers such that*

$$\sigma_{PQ}^2 \leq 256 \left(1 + (\lambda^*)^2 + \rho^2\right) \hat{G}_\theta^2, \text{ and } c_\eta \leq \min \left\{ \frac{\rho}{2\sqrt{2}\hat{G}_\lambda}, \frac{\rho}{16\sqrt{6}\sigma_{PQ}\sqrt{\log 2/\tau}}, \frac{\rho}{C_{PQ}} \right\},$$

*where* $C_{PQ} = 2m_1(1 + 2\kappa)\left(\frac{1}{m_1^2}\|\nabla \hat{R}_Q(\theta_{Q,0})\|^2 + \|\theta_{Q,0}\|^2\right) + \frac{6\sigma_Q^2 \log(2/\tau)}{m_1}$, *for* $\sigma_Q^2 = \left(\frac{\log(T+2\kappa)}{m_1} \sup_{(x,y)\in S_Q} \|\nabla \ell(\theta_{Q,0}; x, y)\|\right)^2$.

*Assume the parameters in Algorithm 3 are set as* $\eta = \frac{c_\eta}{\sqrt{T}}$, $\gamma = \frac{\hat{G}_\theta^2 c_\eta}{\sqrt{T}}$, $\alpha_t = \frac{1}{m_1} \cdot \frac{1}{t+2\kappa}$ *and* $\epsilon_0 = \frac{C_{PQ}}{T+2\kappa_Q}$. *Assume* $T \geq \max\{\frac{4C_{PQ}m_1}{r(3\epsilon_Q)}, (\frac{C_{PQ}\cdot(\log(T+2\kappa_Q))^2}{\hat{G}_\lambda\sqrt{\epsilon_Q \log 1/\tau}})^2\}$, *and* $r(3\epsilon_Q) \leq 1$, *then, with probability at least* $1 - \tau$ *over the randomness of Algorithm 3, the empirical risk of the returned solution satisfies:*

$$\hat{R}_P(\hat{\theta}_{PQ}) - \hat{R}_P(\tilde{\theta}_{PQ}) \lesssim \left(\hat{G}_\theta + \hat{G}_\lambda \sqrt{\log \frac{1}{\tau}}\right) \cdot \left(\frac{\frac{\hat{G}_\theta^2}{c_\eta} + \hat{G}_\theta \hat{G}_\lambda \sqrt{\log \frac{1}{\tau}}}{r(3\epsilon_Q)\sqrt{T}} + \frac{\lambda^* \hat{G}_\lambda \sqrt{\log \frac{1}{\tau}} + \frac{\rho^2}{c_\eta}}{\sqrt{T}}\right).$$

**Statistical Implications.** In this section we give an example of how the above Theorem 4 can be converted generically to statistical guarantees on transfer (as was done for the linear regression case).

For intuition, Theorem 4 guarantees that $\hat{R}_P(\hat{\theta}_{PQ})$ is small, i.e., close to $\hat{R}_P(\tilde{\theta}_{PQ})$, while we also know $\hat{R}_Q(\hat{\theta}_{PQ})$ is also small, i.e., close to $\hat{R}_Q(\hat{\theta}_Q)$ (since $\hat{R}_P(\hat{\theta}_{PQ})$ is a projection of the average iterate onto the constraint set centered at $\hat{\theta}_{Q,T}$). Thus, if in addition, we have concentration of empirical risks to true risks, we can convert the guarantees of Theorem 4 to statistical transfer guarantees.

The results below illustrate the above intuition. These results are expressed in terms of generic relations between $P$ and $Q$ risks given as follows.

**Definition 6** (Weak Modulus [5]). *Given $\epsilon > 0$, we define the modulus*

$$\delta(\epsilon) \doteq \sup \left\{\mathcal{E}_Q(\theta) : \mathcal{E}_P(\theta) \leq \epsilon, \theta \in \Theta\right\}.$$

In words, the weak modulus captures the best achievable $Q$ risk, if the learner only has access to $P$'s data. For instance, in linear regression, as explained in the main paper and shown in [5] it can be upper bounded as $\delta(\epsilon) \leq 2\lambda_{\max}(\Sigma_P^{-1}\Sigma_Q) \cdot \epsilon + 2\mathcal{E}_Q(\theta_P^*)$.

We next consider some traditional concentration results for bounded losses in terms of the Rademacher complexity of the loss class.

**Assumption 4** (Boundedness of Loss). *We assume $\ell(\theta; x, y) \leq M_\ell$ for any $\theta \in \Theta$, $(x, y) \in \mathcal{X} \times \mathcal{Y}$.*

**Remark 3.** *The Assumption 4 hold for a strongly convex loss given that the parameter $\theta$'s norm is bounded, e.g., $\|\theta\| \leq B$, $\forall \theta \in \Theta$. For example, in linear regression, if we assume the label space $\mathcal{Y} \subseteq [-M_y, M_y]$, then $\ell(\theta; x, y) = (\theta^\top x - y)^2 \leq 2B^2 M_x^2 + 2M_y^2$.*

We then introduce the Rademacher complexity which characterizes the complexity of a class and will be used to derive uniform convergence result.

**Definition 7** (Rademacher complexity). *Let $\mathcal{F}$ be a family of functions mapping from $\mathcal{Z}$ to $\mathbb{R}$ and $Z = \{z_1, \ldots, z_n\}$ be the i.i.d. samples drawn from distribution $\mu$. Then, the empirical Rademacher complexity of $\mathcal{F}$ with respect to dataset $Z$ is defined as*

$$\hat{\mathcal{R}}_n(\mathcal{F}) \doteq \mathbb{E}_{\boldsymbol{\varepsilon} \in \{\pm 1\}^n} \left[\sup_{f \in \mathcal{F}} \frac{1}{n} \sum_{i=1}^n \varepsilon_i f(z_i)\right],$$

*where $\varepsilon_1, \ldots, \varepsilon_n$ are i.i.d. Rademacher random variables with $\mathbb{P}\{\varepsilon_i = 1\} = \mathbb{P}\{\varepsilon_i = -1\} = 1/2$. Then Rademacher complexity $\mathcal{R}_n(\mathcal{F})$ is defined as $\mathcal{R}_n(\mathcal{F}) \doteq \mathbb{E}\hat{\mathcal{R}}_n(\mathcal{F})$*

**Assumption 5** (Bounded Rademacher Complexity of Loss Class). *We assume $\mathcal{R}_n(\ell \circ \mathcal{H}) \leq \frac{B_{\mathcal{H},\ell}}{\sqrt{n}}$ for some positive real number $B_{\mathcal{H},\ell}$ which characterizes the complexity of the loss class $\ell \circ \mathcal{H}$.*

**Remark 4.** *The Assumption 5 is standard. Here we give some examples.*

*For linear classifier class $\mathcal{H} \doteq \{x \mapsto \theta^\top x : \theta \in \mathbb{R}^d, \|\theta\| \leq B\}$ with L-Lipschitz loss, the bound [24, Lemma 26.10] is given by $\mathcal{R}_n(\ell \circ \mathcal{H}) \leq \frac{LBM_x}{\sqrt{n}}$.*

*For $l$ layer neural network class $\mathcal{H} \doteq \{x \mapsto W_l\sigma(W_{l-1}\ldots\sigma(W_1 x)) : \|W_j\|_F \leq B_j\}$ with L-Lipschitz loss, the bound [25, Theorem 1] is given by*

$$\mathcal{R}_n(\ell \circ \mathcal{H}) \lesssim \frac{LM_x\sqrt{l}\prod_{j=1}^l B_j}{\sqrt{n}}.$$

*For general VC class $\mathcal{H} \subseteq \{-1, 1\}^{\mathcal{X}}$ with VC dimension $d$, the bound [26, Corollary 3.8] is given by $\mathcal{R}_n(\ell \circ \mathcal{H}) \lesssim \sqrt{\frac{d\log n}{n}}$.*

With the above two assumptions we can derive the following rate of uniform convergence.

**Proposition 1** (Uniform Convergence). *Let $\mu$ denote either $P$ or $Q$. With probability at least $1 - \tau$, the following statement holds:*

$$\sup_{h \in \mathcal{H}} |R_\mu(\theta) - \hat{R}_\mu(\theta)| \leq 2\frac{B_{\mathcal{H},\ell}}{\sqrt{n_\mu}} + M_\ell\sqrt{\frac{\log\frac{2}{\tau}}{2n_\mu}}.$$

**Corollary 1** (Statistical Transfer Guarantees). *Let*

$$\epsilon_P = 2\frac{B_{\mathcal{H},\ell}}{\sqrt{n_P}} + M_\ell\sqrt{\frac{\log\frac{2}{\tau}}{2n_P}}, \epsilon_Q = 2\frac{B_{\mathcal{H},\ell}}{\sqrt{n_Q}} + M_\ell\sqrt{\frac{\log\frac{2}{\tau}}{2n_Q}}.$$

*Then with probability at least $1 - 3\tau$ over the randomness of Algorithm 3 and $S_P$ and $S_Q$, the returned solution satisfies*

$$\mathcal{E}_Q(\hat{\theta}_{PQ}) \leq 5 \cdot \min\{\epsilon_Q, \delta(3\epsilon_P)\},$$

*provided a number of iterations*

$$T \gtrsim \left(\hat{G}_\theta + \hat{G}_\lambda\sqrt{\log\frac{1}{\tau}}\right)^2 \cdot \left(\frac{\frac{\hat{G}_\theta^2}{c_\eta} + \hat{G}_\theta\hat{G}_\lambda\sqrt{\log\frac{1}{\tau}}}{r(2\epsilon_Q)\epsilon_P} + \frac{\lambda^*\hat{G}_\lambda\sqrt{\log\frac{1}{\tau}} + \frac{\rho^2}{c_\eta}}{\epsilon_P}\right)^2.$$

Here we achieve a transfer guarantee similar to that in Theorem 1. The rate is still adaptive—it achieves the better rate between solely target ERM and source ERM. The required iteration number depends on the $\epsilon_P$ and $r(\epsilon_Q)$, also similar to that in Theorem 1.

## 10    Additional Experiments

In this section we provide additional experimental results.

### 10.1    More Results on CIFAR10 Dog vs Cat dataset

Here we provide more results on CIFAR10. To verify the adaptivity of the algorithm with increasing target samples, we conduct the experiment with fixed $n_P = 100$ and gradually increasing $n_Q$ from 100 to 1500. We use SGD with stepsize $10^{-5}$ and epoch number 1000 to compute $P$ and $Q$ ERM model and HTL model. We choose stepsize for $\theta$ to be $10^{-5}$ and that for $\lambda$ to be $10^{-3}$ , as well as $T = 2000 \cdot \max\{n_P, n_Q\}$ in our method. We set the source dataset to be 80% dog samples and 20% cat samples, and the ratio for target is adjusted to 50% dog and 50% cat. As we can see from Figure 6, in this setting source data are not informative so the source ERM performs poorly. As the target sample size increases, the target ERM can give promising performance, and our method can also adapt to it. HTL performs the best in this setting, but its runtime dramatically increases as the target sample size increases.

## 10.2   Binary Classification Results on the Malware IoT dataset

Here we provide results on the CIC-IDS-2017 dataset [27]. It is a network traffic dataset where the goal is to predict whether the traffic is benign (normal) or malicious (abnormal). We train a linear classifier with hinge loss. We use the network traffic data collected on Wednesday (Wednesday-workingHours.pcap_ISCX.csv) as the source dataset, and that on Friday (Friday-WorkingHours-Afternoon-DDos.pcap_ISCX.csv) as the target. We first fix the total $Q$ sample size $n_Q = 1000$ and gradually increase the $P$ sample size from 1000 to 3000. We set stepsize for $\theta$ to be $10^{-5}$ and for $\lambda$ to be $10^{-3}$ and $\epsilon_Q = \frac{1}{\sqrt{n_Q}}$. Since the abnormal data are very few, to ensure an acceptable Type-II error, we use weighted hinge loss with weight $\frac{1}{2}$, i.e., $\hat{R}_\mu(\theta) \doteq \frac{1}{2}\hat{R}_{\mu,1}(\theta) + \frac{1}{2}\hat{R}_{\mu,0}(\theta)$ where $\hat{R}_{\mu,0}(\theta), \hat{R}_{\mu,1}(\theta)$ denote the empirical risk on the normal data and abnormal data respectively, which is widely used in weighted SVM for imbalanced dataset [28]. The results are shown in Figure 7. As we can see, as the number of abnormal data from $P$ is growing, our method gains from the source data and yields a better Type-I and Type-II error than baselines. The runtime of our method is significantly less than HTL since HTL needs expensive cross-validation process.

## 11   Missing Proofs in Section 4.1

In this section, we provide the missing proofs in Section 4.1. We first introduce the following helper lemma that lower bounds the norm of the constraint gradient on the boundard of the constraint set.

**Lemma 8** (Lower and upper bound of boundary gradient). *For any $\epsilon > 0$, the following statements hold:*

$$\min_{\theta:\hat{R}_Q(\theta)-\hat{R}_Q(\hat{\theta}_Q)=\epsilon} \left\|\nabla\hat{R}_Q(\theta)\right\|^2 = \epsilon\lambda^+_{\min}(\hat{\Sigma}_Q).$$

*Proof.* We start by proving the first statement. First, the $\theta$ on the boundary of the constraint set satisfies: $\left\|\theta - \hat{\theta}_Q\right\|^2_{\hat{\Sigma}_Q} = \epsilon$. We first decompose $\theta$ as $\theta = \theta' + \theta^\perp$ where $\theta' \in \text{Range}(\hat{\Sigma}_Q)$ and $\theta^\perp$ is in the null space of $\hat{\Sigma}_Q$ .

Now we examine the gradient norm:

$$\min_{\theta:\hat{R}_Q(\theta)-\hat{R}_Q(\hat{\theta}_Q)=\epsilon} \left\|\nabla\hat{R}_Q(\theta)\right\|^2 = \min_{\left\|\theta-\hat{\theta}_Q\right\|^2_{\hat{\Sigma}_Q}=\epsilon} \left\|\hat{\Sigma}_Q(\theta-\hat{\theta}_Q)\right\|^2$$

$$= \min_{\left\|\theta'-\hat{\theta}_Q\right\|^2_{\hat{\Sigma}_Q}=\epsilon, \theta'\in\text{Range}(\hat{\Sigma}_Q)} \left\|\hat{\Sigma}_Q(\theta'-\hat{\theta}_Q)\right\|^2$$

$$= \min_{\mathbf{u}\in\text{Range}(\hat{\Sigma}_Q),\|\mathbf{u}\|=1} \left\|\sqrt{\epsilon}\hat{\Sigma}_Q^{\frac{1}{2}}\mathbf{u}\right\|^2 = \epsilon\lambda^+_{\min}(\hat{\Sigma}_Q).$$

$\square$

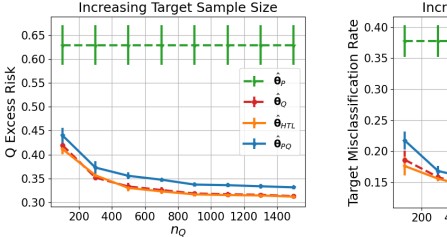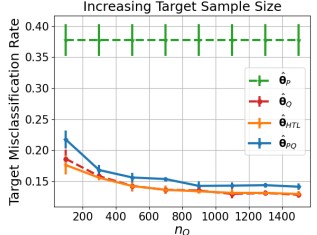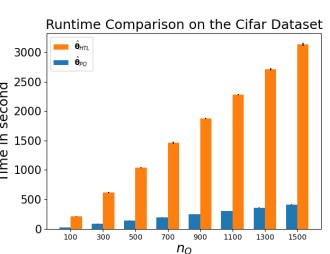

Figure 6:   Binary Classification on the CIFAR10 Dog vs Cat dataset. We fix $n_P = 100$ and increase $n_Q$ gradually. (Left) Target excess risk (Middle) Target misclassification rate. (Right) Runtime comparison. Our method ($\hat{\theta}_{PQ}$) is still comparable with target ERM. HTL ($\hat{\theta}_{\text{HTL}}$) slightly outperforms ours, but as $n_Q$ increases, the runtime of HTL dramatically increases.

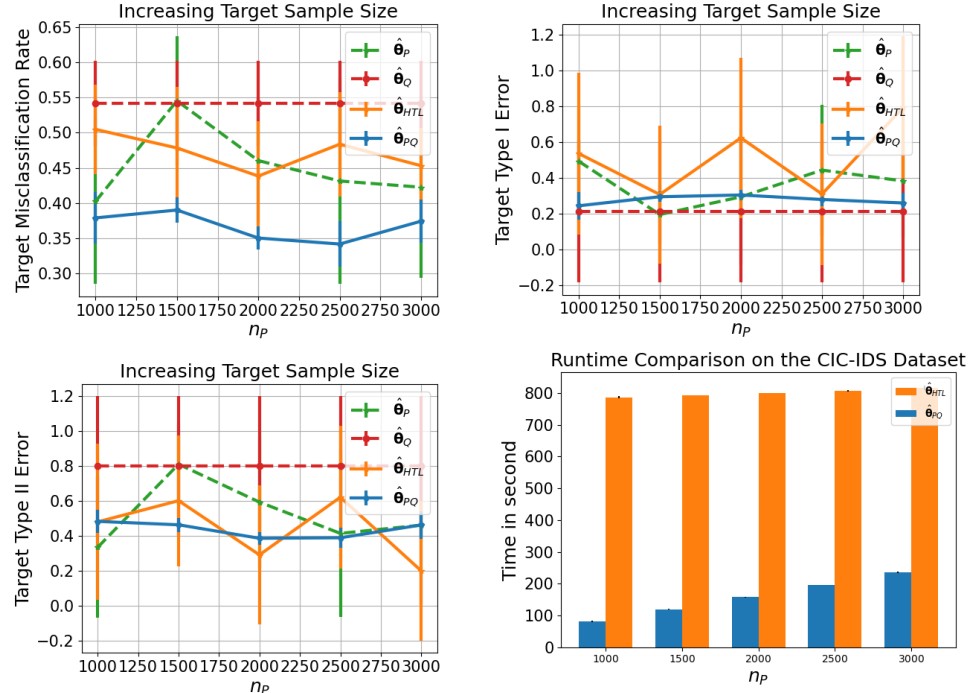

Figure 7: Binary Classification on the CIC-IDS-2017 dataset. We fix $n_Q = 1000$ (total samples from $Q$) and increase $n_P$ (abnormal samples) gradually. (Top-Left) Overall Target Error (Top-Right) Target Type-I Error (Bottom-Left) Target Type-II Error (Bottom-Right) Runtime comparison. Our method ($\hat{\theta}_{PQ}$) achieves better overall accuracy than baselines. We also achieve better and more stable Type-II error than all other baselines. The runtime of our method is also significantly less than HTL.

**Proof of Lemma 1:**

*Proof.* Due to Jensen's inequality, we have that $\left\| \theta_0 - \tilde{\theta}_{PQ} \right\|^2 \leq 2 \left\| \theta_0 - \hat{\theta}_P \right\|^2 + 2 \left\| \hat{\theta}_P - \tilde{\theta}_{PQ} \right\|^2$, which is at most

$$\frac{1}{2\lambda_{\min}^2(\hat{\Sigma}_P)} \left\| \nabla \hat{R}_P(\theta_0) \right\|^2 + 2 \left\| \hat{\theta}_P - \tilde{\theta}_{PQ} \right\|^2 \leq \frac{\left\| \nabla \hat{R}_P(\theta_0) \right\|^2}{2\lambda_{\min}^2(\hat{\Sigma}_P)} + \frac{2 \left\| \hat{\theta}_P - \tilde{\theta}_{PQ} \right\|_{\hat{\Sigma}_P}^2}{\lambda_{\min}(\hat{\Sigma}_P)}.$$

Since $\tilde{\theta}_{PQ}$ is the minimizer of $\hat{R}_P$ within constraint set, and $\hat{\theta}_Q$ is also in the constraint set, we have

$$\left\| \hat{\theta}_P - \tilde{\theta}_{PQ} \right\|_{\hat{\Sigma}_P}^2 \leq \left\| \hat{\theta}_P - \hat{\theta}_Q \right\|_{\hat{\Sigma}_P}^2 \leq \frac{\left\| \nabla \hat{R}_P(\hat{\theta}_Q) \right\|^2}{4\lambda_{\min}(\hat{\Sigma}_P)}.$$

Putting pieces together we have

$$\left\| \theta_0 - \tilde{\theta}_{PQ} \right\|^2 \leq \frac{\left\| \nabla \hat{R}_P(\theta_0) \right\|^2}{2\lambda_{\min}^2(\hat{\Sigma}_P)} + \frac{\left\| \nabla \hat{R}_P(\hat{\theta}_Q) \right\|^2}{2\lambda_{\min}^2(\hat{\Sigma}_P)}.$$

□

**Proof of Lemma 2:**

*Proof.* For $\theta$ such that $\left\| \theta - \tilde{\theta}_{PQ} \right\|^2 \leq 2\rho^2$, we examine the upper bound of its norm. According to triangle inequality we have:

$$\|\theta\| \leq \sqrt{2}\rho + \left\| \tilde{\theta}_{PQ} \right\| \leq 3\rho.$$

The rest of the proof follows:

$$\|\nabla \ell(\theta; x, y)\| = \left\|x(\theta^\top x - y)\right\| \le M_x^2 \|\theta\| + M_x \hat{M}_y.$$

$\square$

**Proof of Lemma 3:**

*Proof.* The proof simply follows the definition of $\ell$:

$$
\begin{aligned}
|\ell(\theta; x, y) - \ell(\theta'; x, y) - 6\epsilon_Q| &\le (\theta^\top x - y)^2 + (\theta'^\top x - y)^2 + 6\epsilon_Q \\
&\le 2M_x^2 \|\theta\|^2 + 2\hat{M}_y + 2M_x^2 \|\theta'\|^2 + \epsilon_Q \\
&\le 18 \left( M_x^2 \rho^2 + \hat{M}_y + \frac{M_x^4 \hat{M}_y^2 (1 + \log(T + 2\kappa_Q))^2}{(\lambda_{\min}^+(\hat{\Sigma}_Q))^2} \right) + 6\epsilon_Q.
\end{aligned}
$$

$\square$

**Proof of Lemma 4:**

*Proof.* Due to the first order optimality condition we have

$$\nabla \hat{R}_P(\tilde{\theta}_{PQ}) + \lambda^* \nabla \hat{R}_Q(\tilde{\theta}_{PQ}) = 0 \implies \lambda^* = \frac{\left\|\nabla \hat{R}_P(\tilde{\theta}_{PQ})\right\|}{\left\|\nabla \hat{R}_Q(\tilde{\theta}_{PQ})\right\|}.$$

To upper bound $\left\|\nabla \hat{R}_P(\tilde{\theta}_{PQ})\right\|$ we notice that

$$
\begin{aligned}
\left\|\nabla \hat{R}_P(\tilde{\theta}_{PQ})\right\| &= \left\|2\hat{\Sigma}_P(\tilde{\theta}_{PQ} - \hat{\theta}_P)\right\| \\
&\le 2\sqrt{\lambda_{\max}(\hat{\Sigma}_P)} \left\|\hat{\Sigma}_P^{1/2}(\tilde{\theta}_{PQ} - \hat{\theta}_P)\right\| = 2\sqrt{\lambda_{\max}(\hat{\Sigma}_P)} \sqrt{\hat{R}_P(\tilde{\theta}_{PQ}) - \hat{R}_P(\hat{\theta}_P)}.
\end{aligned}
$$

Define $\theta'$ to be the model on the boundary of constraint set, and also on the range of $\hat{\Sigma}_Q$. That is, $\left\|\theta' - \hat{\theta}_Q\right\|_{\hat{\Sigma}_Q}^2 = 6\epsilon_Q$ and $\theta' \in \text{Range}(\hat{\Sigma}_Q)$. Since $\tilde{\theta}_{PQ}$ is the minimizer of $\hat{R}_P(\cdot)$ in the constraint set, we know that

$$
\begin{aligned}
\left\|\nabla \hat{R}_P(\tilde{\theta}_{PQ})\right\| &\le 2\sqrt{\lambda_{\max}(\hat{\Sigma}_P)} \sqrt{\hat{R}_P(\theta') - \hat{R}_P(\hat{\theta}_P)} \\
&= 2\sqrt{\lambda_{\max}(\hat{\Sigma}_P)} \left\|\hat{\Sigma}_P^{1/2}(\theta' - \hat{\theta}_P)\right\| \\
&\le 2\lambda_{\max}(\hat{\Sigma}_P) \left\|\theta' - \hat{\theta}_P\right\| \\
&\le 2\lambda_{\max}(\hat{\Sigma}_P) \left\|\theta' - \hat{\theta}_Q\right\| + 2\lambda_{\max}(\hat{\Sigma}_P) \left\|\hat{\theta}_Q - \hat{\theta}_P\right\| \\
&\le 2\lambda_{\max}(\hat{\Sigma}_P) \frac{1}{\sqrt{\lambda_{\min}^+(\hat{\Sigma}_Q)}} \left\|\hat{\Sigma}_Q^{1/2}(\theta' - \hat{\theta}_Q)\right\| + 2\lambda_{\max}(\hat{\Sigma}_P) \left\|\hat{\theta}_Q - \hat{\theta}_P\right\| \\
&= \frac{2\lambda_{\max}(\hat{\Sigma}_P)}{\sqrt{\lambda_{\min}^+(\hat{\Sigma}_Q)}} \cdot \sqrt{6\epsilon_Q} + 2\lambda_{\max}(\hat{\Sigma}_P) \left\|\hat{\theta}_Q - \hat{\theta}_P\right\|.
\end{aligned}
$$

To lower bound $\left\|\nabla \hat{R}_Q(\tilde{\theta}_{PQ})\right\|$, we again evoke Lemma 8 that $\left\|\nabla \hat{R}_Q(\tilde{\theta}_{PQ})\right\| \ge 2\sqrt{\lambda_{\min}^+(\hat{\Sigma}_Q) \cdot 6\epsilon_Q}$. Putting pieces together will conclude the proof.

$\square$

## 12 Missing Proofs in Section 5.2

**Proof of Lemma 5:**

*Proof.* The proof mainly follows Proposition 8 of [5]. With probability at least $1 - 2\tau$, the following two facts hold. For one hand, for any $\theta \in \left\{ \theta : \left\| \theta - \hat{\theta}_Q \right\|_{\hat{\Sigma}_Q}^2 \leq 6\epsilon_Q \right\}$, we know

$$\mathcal{E}_Q(\theta) = \left\| \theta - \theta_Q^* \right\|_{\Sigma_Q}^2 \leq 2 \left\| \theta - \hat{\theta}_Q \right\|_{\Sigma_Q}^2 + 2 \left\| \theta_Q^* - \hat{\theta}_Q \right\|_{\Sigma_Q}^2 \leq 4 \left\| \theta - \hat{\theta}_Q \right\|_{\hat{\Sigma}_Q}^2 + 2\epsilon_Q \leq 26\epsilon_Q.$$

where at the second inequality we evoke the matrix concentration result from Lemma 3 of [5]. For the other hand, for any $\theta$ such that $\mathcal{E}_Q(\theta) \leq \epsilon_Q$, we have

$$\left\| \theta - \hat{\theta}_Q \right\|_{\hat{\Sigma}_Q}^2 \leq \frac{3}{2} \left\| \theta - \hat{\theta}_Q \right\|_{\Sigma_Q}^2 \leq 3 \left\| \theta - \theta_Q^* \right\|_{\Sigma_Q}^2 + 3 \left\| \hat{\theta}_Q - \theta_Q^* \right\|_{\Sigma_Q}^2 \leq 6\epsilon_Q.$$

$\square$

**Proof of Lemma 6:**

*Proof.* First notice the following decomposition: $\| \theta - \theta_P^* \|_{\Sigma_P}^2 \leq 2 \left\| \theta - \tilde{\theta}_{PQ} \right\|_{\Sigma_P}^2 + 2 \left\| \tilde{\theta}_{PQ} - \theta_P^* \right\|_{\Sigma_P}^2$. For the first term in RHS of above inequality, with probability at least $1 - \tau$, we have

$$2 \left\| \theta - \tilde{\theta}_{PQ} \right\|_{\Sigma_P}^2 \leq 4 \left\| \theta - \tilde{\theta}_{PQ} \right\|_{\hat{\Sigma}_P}^2 = 4 \left( \hat{R}_P(\theta) - \hat{R}_P(\tilde{\theta}_{PQ}) - \left\langle \nabla \hat{R}_P(\tilde{\theta}_{PQ}), \theta - \tilde{\theta}_{PQ} \right\rangle \right).$$

Since both $\theta$ and $\tilde{\theta}_{PQ}$ are in the constraint set of Problem (2), and $\tilde{\theta}_{PQ}$ is the optimal solution within the set, we know $\left\langle \nabla \hat{R}_P(\tilde{\theta}_{PQ}), \theta - \tilde{\theta}_{PQ} \right\rangle \geq 0$. Hence, we know $2 \left\| \theta - \tilde{\theta}_{PQ} \right\|_{\Sigma_P}^2 \leq 4 \left( \hat{R}_P(\theta) - \hat{R}_P(\tilde{\theta}_{PQ}) \right)$.

Now we proceed to bounding $2 \left\| \tilde{\theta}_{PQ} - \theta_P^* \right\|_{\Sigma_P}^2$. Notice that with probability at least $1 - 2\tau$

$$2 \left\| \tilde{\theta}_{PQ} - \theta_P^* \right\|_{\Sigma_P}^2 \leq 4 \left\| \tilde{\theta}_{PQ} - \hat{\theta}_P \right\|_{\Sigma_P}^2 + 4 \left\| \hat{\theta}_P - \theta_P^* \right\|_{\Sigma_P}^2$$

$$\leq 8 \left\| \tilde{\theta}_{PQ} - \hat{\theta}_P \right\|_{\hat{\Sigma}_P}^2 + 4 \left\| \hat{\theta}_P - \theta_P^* \right\|_{\Sigma_P}^2$$

$$\leq 8 \left\| \theta_P^* - \hat{\theta}_P \right\|_{\hat{\Sigma}_P}^2 + 4 \left\| \hat{\theta}_P - \theta_P^* \right\|_{\Sigma_P}^2$$

$$\leq 12 \left\| \theta_P^* - \hat{\theta}_P \right\|_{\Sigma_P}^2 + 4 \left\| \hat{\theta}_P - \theta_P^* \right\|_{\Sigma_P}^2$$

$$\leq 16\epsilon_P,$$

where at second and fourth step we evoke matrix concentration result from Lemma 3 of [5], at third step we use the fact that $\tilde{\theta}_{PQ}$ is the optimal solution within the set. Putting pieces together will conclude the proof. $\square$

**Proof of Lemma 7:**

*Proof.* Since $\theta_P^* \notin \left\{ \theta : \left\| \theta - \hat{\theta}_Q \right\|_{\hat{\Sigma}_Q}^2 \leq 6\epsilon_Q \right\}$, then from Lemma 5 we know with probability at least $1 - 2\tau$, $\mathcal{E}_Q(\theta_P^*) \geq \epsilon_Q \geq \frac{1}{26}\mathcal{E}_Q(\theta)$ holds. $\square$

## 13    Proof of Convergence

In this section we will present the proof of the convergence result. We first introduce some useful lemmas.

### 13.1    Technical Lemmas

The following proposition is standard and will be used to show the sub-gaussianity of the stochastic gradients.

**Proposition 2.** *Given a random variable $X$, if $a \leq X \leq b$ with probability 1, then $X$ is a $\frac{(b-a)^2}{4}$ sub-Gaussian random variable.*

The next lemma establishes the convergence of $\theta_{Q,t}$.

**Lemma 9** (High probability convergence of $\theta_{Q,t}$). *If we choose $\alpha_t = \frac{1}{\lambda_{\min}^+(\hat{\Sigma}_Q)(t+2\kappa_Q)}$, then with probability at least $1 - \tau$, for any $t \geq 0$ we have:*

$$\hat{R}_Q(\theta_{Q,t}) - \hat{R}_Q(\hat{\theta}_Q) \leq \frac{\lambda_{\min}^+(\hat{\Sigma}_Q)(1+2\kappa_Q)\left\|\hat{\theta}_Q\right\|^2}{t+2\kappa_Q} + \frac{6\sigma_Q^2 \log(2/\tau)(\log t + 1)}{\lambda_{\min}^+(\hat{\Sigma}_Q)(t+2\kappa_Q)}$$

*for $\sigma_Q^2 = \left(M_x^2\left(\frac{1+\log(T+2\kappa_Q)}{\lambda_{\min}^+(\hat{\Sigma}_Q)}M_xM_y\right) + M_xM_y\right)^2$.*

*Proof.* We first examine the boundedness of $\|\theta_{Q,t}\|$. According to updating rule of $\theta_{Q,t}$ we have

$$\begin{aligned}
\|\theta_{Q,t}\| &= \left\|\theta_{Q,t-1} - \alpha_t x_t(x_t^\top \theta_{Q,t-1} - y_t)\right\| \\
&\leq \left\|(\mathbf{I} - \alpha_t x_t x_t^\top)\theta_{Q,t-1}\right\| + \alpha_t \|x_t y_t\| \\
&\leq \underbrace{\|\theta_{Q,0}\|}_{=0} + \sum_{s=1}^t \frac{1}{\lambda_{\min}^+(\hat{\Sigma}_Q)(s+2\kappa_Q)}M_x\hat{M}_y \\
&\leq \frac{1 + \log(t+2\kappa_Q)}{\lambda_{\min}^+(\hat{\Sigma}_Q)}M_xM_y.
\end{aligned} \tag{8}$$

Hence we can compute the sub-Gaussian parameter. Notice that

$$\begin{aligned}
\left\|\nabla\ell(\theta_{Q,t};x_t,y_t) - \nabla\hat{R}_Q(\theta_{Q,t})\right\| &\leq \left\|x_t x_t^\top - \hat{\Sigma}_Q\right\|\|\theta_{Q,t}\| + \left\|x_t y_t - \frac{1}{n_Q}\mathbf{X}_Q^\top \mathbf{y}_Q\right\| \\
&\leq 2M_x^2\left(\frac{1+\log(t+2\kappa_Q)}{\lambda_{\min}^+(\hat{\Sigma}_Q)}M_xM_y\right) + 2M_x\hat{M}_y \\
&\leq 2M_x^2\left(\frac{1+\log(T+2\kappa_Q)}{\lambda_{\min}^+(\hat{\Sigma}_Q)}M_xM_y\right) + 2M_x\hat{M}_y.
\end{aligned}$$

According to Proposition 2, we know $\left\|\nabla\ell(\theta_{Q,t};x_t,y_t) - \nabla\hat{R}_Q(\theta_{Q,t})\right\|$ is $\left(M_x^2\left(\frac{1+\log(t+2\kappa_Q)}{\lambda_{\min}^+(\hat{\Sigma}_Q)}M_xM_y\right) + M_xM_y\right)^2$ sub-Gaussian.

Now, we evoke the result from Theorem 3.7 from [29] that if the gradient noise is $\sigma_Q$ sub-Gaussian, then with our choice of $\alpha_t$, with probability at least $1 - \tau$ it holds for any integer $t > 0$ that

$$\hat{R}_Q(\theta_{Q,t}) - \hat{R}_Q(\hat{\theta}_Q) \leq \frac{\lambda_{\min}^+(\hat{\Sigma}_Q)(1+2\kappa_Q)\left\|\hat{\theta}_Q\right\|^2}{t+2\kappa_Q} + \frac{6\sigma_Q^2 \log(2/\tau)(\log t + 1)}{\lambda_{\min}^+(\hat{\Sigma}_Q)(t+2\kappa_Q)}.$$

$\square$

The next lemma proves the sub-Gaussianity of the stochastic gradient used to update $\theta_t$.

**Lemma 10.** *Let*

$$g_\theta = \begin{cases} (1+\lambda)\nabla\ell(\theta; x, y), (x, y) \sim S_P, w.p. \ \frac{1}{1+\lambda} \\ (1+\lambda)\nabla\ell(\theta; x, y), (x, y) \sim S_Q, w.p. \ \frac{\lambda}{1+\lambda} \end{cases}$$

*and*

$$\delta = \left\| \nabla\hat{R}_P(\theta) + \lambda\nabla\hat{R}_Q(\theta) - g_\theta \right\|.$$

*Then for any* $\theta \in \left\{\theta : \|\theta - \tilde{\theta}_{PQ}\|^2 \leq 2\rho^2\right\}$ *and* $\lambda \in \left\{\lambda : (\lambda - \lambda^*)^2 \leq 2\rho^2\right\}$, *we have* $\mathbb{E}[\exp(\delta^2/\sigma_{PQ}^2)] \leq \exp(1)$ *for* $\sigma_{PQ}^2 = 256\left(1 + (\lambda^*)^2 + \rho^2\right)\hat{G}_\theta^2$.

*Proof.* We use $\xi = 1$ to denote the event that we sample from $P$, and otherwise from $Q$. For notational convenience we define $\delta_\mu \doteq \|\nabla\hat{R}_\mu(\theta) - \nabla\ell(\theta; x, y)\|$ where $x, y$ is sampled from $\mu$, for $\mu$ denoting either $P$ or $Q$. We also define $\zeta_{PQ} \doteq \left\|\nabla\hat{R}_P(\theta) - \nabla\hat{R}_Q(\theta)\right\|$. Since $\delta = \left\|\nabla\hat{R}_P(\theta) + \lambda\nabla\hat{R}_Q(\theta) - (1+\lambda)(\mathbb{I}\{\xi = 1\}\nabla\ell(\theta; x, y) + \mathbb{I}\{\xi = 0\}\nabla\ell(\theta; x, y))\right\|$, we can verify that

$$\mathbb{E}\exp\left(\delta^2/\sigma_{PQ}^2\right)$$

$$= \frac{1}{1+\lambda}\mathbb{E}\exp\left(\left\|\nabla\hat{R}_P(\theta) + \lambda\nabla\hat{R}_Q(\theta) - (1+\lambda)\nabla\ell(\theta; x, y)\right\|^2/\sigma_{PQ}^2\right)$$

$$+ \frac{\lambda}{1+\lambda}\mathbb{E}\exp\left(\left\|\nabla\hat{R}_P(\theta) + \lambda\nabla\hat{R}_Q(\theta) - (1+\lambda)\nabla\ell(\theta; x, y)\right\|^2/\sigma_{PQ}^2\right)$$

$$\leq \frac{1}{1+\lambda}\mathbb{E}\exp\left(\frac{2(1+\lambda)^2\delta_P^2}{\sigma_{PQ}^2} + \frac{2\lambda^2\zeta_{PQ}^2}{\sigma_{PQ}^2}\right) + \frac{\lambda}{1+\lambda}\mathbb{E}\exp\left(\frac{2\zeta_{PQ}^2}{\sigma_{PQ}^2} + \frac{2(1+\lambda)^2\delta_Q^2}{\sigma_{PQ}^2}\right)$$

$$\leq \frac{1}{1+\lambda}\left(\exp(\frac{2\lambda^2\zeta_{PQ}^2}{\sigma_{PQ}^2})\mathbb{E}\exp\left(2(1+\lambda)^2\frac{\delta_P^2}{\sigma_{PQ}^2}\right) + \lambda\exp\left(\frac{2\zeta_{PQ}^2}{\sigma_{PQ}^2}\right)\mathbb{E}\exp(\frac{2(1+\lambda)^2\delta_Q^2}{\sigma_{PQ}^2})\right).$$

Since $0 \leq \lambda \leq \sqrt{2\rho} + \lambda^*$, so we have

$$\mathbb{E}\exp\left(\delta^2/\hat{\sigma}^2\right)$$

$$\leq \frac{1}{1+\lambda}\exp\left(2(2(\lambda^*)^2 + 4\rho^2)4\hat{G}_\theta^2/\sigma_{PQ}^2\right)\mathbb{E}\exp\left((4 + 4(2(\lambda^*)^2 + 4\rho^2))\delta_P^2/\sigma_{PQ}^2\right)$$

$$+ \frac{\lambda_t}{1+\lambda_t}\exp\left(4\hat{G}_\theta^2/\sigma_{PQ}^2\right)\mathbb{E}\exp\left((4 + 4(2(\lambda^*)^2 + 4\rho^2))\delta_Q^2/\sigma_{PQ}^2\right).$$

Due to our choice $\sigma_{PQ}^2 = 256\left(1 + (\lambda^*)^2 + \rho^2\right)\hat{G}_\theta^2$, we have

$$\mathbb{E}\exp\left(\delta^2/\sigma_{PQ}^2\right) \leq \frac{1}{1+\lambda}\exp(1/8)(\exp(1))^{1/8} + \frac{\lambda}{1+\lambda}\exp(1/8)(\exp(1))^{1/8} \leq \exp(1).$$

$\square$

Then, we establish the convergence of the penalized objective, under the dynamic of Algorithm 1.

**Lemma 11.** *For Algorithm 1, the following statement holds true for any* $\lambda \geq 0$ *with probability at least* $1 - \tau$:

$$\left(\hat{R}_P(\bar{\theta}_T) - \hat{R}_P(\tilde{\theta}_{PQ})\right) + \lambda(\hat{R}_Q(\bar{\theta}_T) - \hat{R}_Q(\hat{\theta}_Q) - 6\epsilon_Q) - \left(\frac{\gamma}{2} + \frac{1}{2\eta T}\right)\lambda^2$$

$$\leq \frac{\rho^2}{\eta T} + \eta\hat{G}_\lambda^2 + \eta\hat{G}_\theta^2 + \frac{\hat{G}_\lambda\sqrt{3\log\frac{2}{\tau}}}{\sqrt{T}}\left(\lambda^* + 2\sqrt{2}\rho\right) + \frac{\sqrt{2}\rho\sigma_{PQ}\sqrt{3\log\frac{2}{\tau}}}{\sqrt{T}}$$

$$+ \frac{C_{PQ}(\log(T + 2\kappa_Q) + 2)^2\left(\lambda^* + \sqrt{2}\rho\right)}{T} + \left(\frac{\hat{G}_\lambda\sqrt{3\log\frac{2}{\tau}}}{\sqrt{T}} + \frac{C_{PQ}(\log(T + 2\kappa_Q) + 2)^2}{T}\right)\lambda,$$

*where* $C_{PQ} := (1 + 2\kappa_Q)M_x^2 M_y^2 + \frac{6\sigma_Q^2\log(2/\tau)}{\lambda_{\min}^+(\hat{\Sigma}_Q)}$, $\sigma_Q^2 = \left(M_x^2\left(\frac{1+\log(T+2\kappa_Q)}{\lambda_{\min}^+(\hat{\Sigma}_Q)}M_x M_y\right) + M_x M_y\right)^2$.

*Proof.* Define the constraint function $g(\theta) := \hat{R}_Q(\theta) - \hat{R}_Q(\hat{\theta}_Q) - 6\epsilon_Q$ and $L(\theta, \lambda) \doteq \hat{R}_P(\theta) + \lambda g(\theta) - \frac{\gamma\lambda^2}{2}$. We first show that $\left\|\theta_t - \tilde{\theta}_{PQ}\right\|^2 + \|\lambda_t - \lambda^*\|^2 \leq 2\rho^2$, for any $t \in [T]$. We prove this by induction. Assume this holding for $t$, and for $t+1$ we have

$$\left\|\theta_{t+1} - \tilde{\theta}_{PQ}\right\|^2 = \left\|\theta_t - \eta g_\theta^t - \tilde{\theta}_{PQ}\right\|^2$$
$$= \left\|\theta_t - \tilde{\theta}_{PQ}\right\|^2 - 2\left\langle \eta g_\theta^t, \theta_t - \tilde{\theta}_{PQ}\right\rangle + \eta^2 \left\|g_\theta^t\right\|^2,$$

where $g_\theta^t = \begin{cases} (1+\lambda_t)\nabla\ell(\theta_t; x_t, y_t), (x_t, y_t) \sim S_P, w.p. \frac{1}{1+\lambda_t} \\ (1+\lambda_t)\nabla\ell(\theta_t; x_t, y_t), (x_t, y_t) \sim S_Q, w.p. \frac{\lambda_t}{1+\lambda_t} \end{cases}$.

Then for $t+1$, we have:

$$\left\|\theta_{t+1} - \tilde{\theta}_{PQ}\right\|^2 \leq \left\|\theta_t - \tilde{\theta}_{PQ}\right\|^2 - 2\eta_t\left\langle \nabla\hat{R}_P(\theta_t) + \lambda_t\nabla\hat{R}_Q(\theta_t), \theta_t - \tilde{\theta}_{PQ}\right\rangle$$
$$+ 2\sqrt{2}\eta\delta_t\rho + \eta_t^2(1+\lambda_t)^2\hat{G}_\theta^2$$
$$\leq \left\|\theta_t - \tilde{\theta}_{PQ}\right\|^2 - 2\eta\left(L(\theta_t, \lambda_t) - L(\tilde{\theta}_{PQ}, \lambda_t)\right) + 2\sqrt{2}\eta\delta_t\rho + \eta^2(1+\lambda_t)^2\hat{G}_\theta^2$$

where $\delta_t = \left\|\nabla\hat{R}_P(\theta_t) + \lambda_t\nabla\hat{R}_Q(\theta_t) - g_\theta^t\right\|$, and at last step we use the convexity of $L(\cdot, \lambda_t)$. According to Lemma 10 we know that

$$\mathbb{E}[g_\theta^t] = \nabla\hat{R}_P(\theta_t) + \lambda_t\nabla\hat{R}_Q(\theta_t), \mathbb{E}[\exp\left(\delta_t^2/\sigma_{PQ}^2\right)] \leq \exp(1)$$

. Similarly, we have:

$$|\lambda_{t+1} - \lambda^*|^2 = |\lambda_t - \lambda^*|^2 + 2\eta\left\langle g_\lambda^t, \lambda_t - \lambda^*\right\rangle + \eta^2|g_\lambda^t - \gamma\lambda_t|^2$$
$$\leq |\lambda_t - \lambda^*|^2 + 2\eta\left\langle \hat{R}_Q(\theta_t) - \hat{R}_Q(\hat{\theta}_Q) - \epsilon_Q - \gamma\lambda_t, \lambda_t - \lambda^*\right\rangle$$
$$+ 2\sqrt{2}\eta r_t\rho + 2\sqrt{2}\eta h_t\rho + 2\eta^2\hat{G}_\lambda^2$$
$$\leq (1-\gamma\eta)|\lambda_t - \lambda|^2 - 2\eta\left(L(\theta_t, \lambda^*) - L(\theta_t, \lambda_t)\right)$$
$$+ 2\sqrt{2}\eta r_t\rho + 2\sqrt{2}\eta h_t\rho + 2\eta^2\hat{G}_\lambda^2,$$

where

$$g_\lambda^t = \ell(\theta_t; x_t, y_t) - \ell(\theta_{Q,t}; x_t, y_t) - 6\epsilon_Q$$

and

$$r_t = \left\|\ell(\theta_t; x_t, y_t) - \ell(\theta_{Q,t}; x_t, y_t) - (\hat{R}_Q(\theta_t) - \hat{R}_Q(\theta_{Q,t}))\right\|, h_t = |\hat{R}_Q(\theta_{Q,t}) - \hat{R}_Q(\hat{\theta}_Q)|$$

and at last step we use the $\gamma$ concavity of $L(\theta_t, \cdot)$. It is easy to see that

$$\mathbb{E}[g_\lambda^t] = \hat{R}_Q(\theta_t) - \hat{R}_Q(\hat{\theta}_Q) - \epsilon_Q, \mathbb{E}[\exp\left(r_t^2/\hat{G}_\lambda^2\right)] \leq \exp(1).$$

Putting pieces together we have

$$\left\|\theta_{t+1} - \tilde{\theta}_{PQ}\right\|^2 + |\lambda_{t+1} - \lambda^*|^2 \leq \left(|\lambda_t - \lambda^*|^2 + \left\|\theta_t - \tilde{\theta}_{PQ}\right\|^2\right) - 2\eta\left(L(\theta_t, \lambda^*) - L(\tilde{\theta}_{PQ}, \lambda_t)\right)$$
$$+ 2\sqrt{2}\eta\delta_t\rho + 2\sqrt{2}\eta r_t\rho + 2\sqrt{2}\eta h_t\rho + 2\eta^2\hat{G}_\lambda^2 + \eta^2(1+\lambda_t)^2\hat{G}_\theta^2$$
$$\leq \left(|\lambda_t - \lambda^*|^2 + \left\|\theta_t - \tilde{\theta}_{PQ}\right\|^2\right) - 2\eta\underbrace{\left(L(\theta_t, \lambda^*) - L(\tilde{\theta}_{PQ}, \lambda_t)\right)}_{\geq -\frac{\gamma(\lambda^*)^2}{2}}$$
$$+ 2\eta^2\hat{G}_\lambda^2 + \eta^2(1+\sqrt{2}\rho+\lambda^*)^2\hat{G}_\theta^2 + 2\sqrt{2}\eta\rho(\delta_t + r_t + h_t)$$

where the last step is due to

$$L(\theta_t, \lambda^*) - L(\tilde{\theta}_{PQ}, \lambda_t) = \underbrace{\hat{R}_P(\theta_t) + \lambda^* g(\theta_t) - \left(\hat{R}_P(\tilde{\theta}_{PQ}) + \lambda_t g(\tilde{\theta}_{PQ})\right)}_{\geq 0} - \frac{\gamma(\lambda^*)^2}{2} + \frac{\gamma\lambda_t^2}{2}.$$

Performing telescoping sum yields:

$$\left\|\theta_{t+1} - \tilde{\theta}_{PQ}\right\|^2 + |\lambda_{t+1} - \lambda^*|^2 \leq \left(\left\|\theta_0 - \tilde{\theta}_{PQ}\right\|^2 + |\lambda_0 - \lambda^*|^2\right) + 2\eta^2\hat{G}_\lambda^2 + \eta^2(1 + \sqrt{2}\rho + \lambda^*)^2\hat{G}_\theta^2$$

$$+ 2\sqrt{2}\eta\rho\sum_{s=0}^{t}\delta_s + 2\sqrt{2}\eta\rho\sum_{s=0}^{t}r_s + 2\sqrt{2}\eta\rho\sum_{s=0}^{t}h_s + T\gamma\eta(\lambda^*)^2.$$

Due to Lemma 4 of [30], we know with probability $1 - \tau/2$,

$$\sum_{t=0}^{T-1}\delta_t \leq \sqrt{T\sigma_{PQ}^2}\sqrt{3\log\frac{2}{\tau}}, \sum_{t=0}^{T-1}r_t \leq \sqrt{T\hat{G}_\lambda^2}\sqrt{3\log\frac{2}{\tau}}, \tag{9}$$

and also according to Lemma 9, $h_t \leq \frac{\lambda_{\min}^+(\hat{\Sigma}_Q)(1+2\kappa_Q)\|\hat{\theta}_Q\|^2}{t+2\kappa_Q} + \frac{6\sigma_Q^2\log(2/\tau)(\log t+1)}{\lambda_{\min}^+(\hat{\Sigma}_Q)(t+2\kappa_Q)}$ for $\sigma_Q^2 = \left(M_x^2\left(\frac{1+\log(T+2\kappa_Q)}{\lambda_{\min}^+(\hat{\Sigma}_Q)}M_xM_y\right) + M_xM_y\right)^2$, which yields:

$$\sum_{s=0}^{t}h_s = \sum_{s=0}^{t}\left(\frac{\lambda_{\min}^+(\hat{\Sigma}_Q)(1+2\kappa_Q)\left\|\hat{\theta}_Q\right\|^2}{s+2\kappa_Q} + \frac{6\sigma_Q^2\log(2/\tau)(\log s+1)}{\lambda_{\min}^+(\hat{\Sigma}_Q)(s+2\kappa_Q)}\right)$$

$$\leq \left(\lambda_{\min}^+(\hat{\Sigma}_Q)(1+2\kappa_Q)\left\|\hat{\theta}_Q\right\|^2 + \frac{6\sigma_Q^2\log(2/\tau)(\log t+1)}{\lambda_{\min}^+(\hat{\Sigma}_Q)}\right)(\log(t+2\kappa_Q)+2)$$

$$\leq C_{PQ}\cdot(\log(t+2\kappa_Q)+2)^2, \tag{10}$$

where $C_{PQ} := (1+2\kappa_Q)M_x^2M_y^2 + \frac{6\sigma_Q^2\log(2/\tau)}{\lambda_{\min}^+(\hat{\Sigma}_Q)}$, and in the first inequality we use the fact $\sum_{s=1}^{t}\frac{1}{s} \leq 1 + \int_1^t\frac{1}{s} \leq 1 + \log t$. Putting pieces together yields:

$$\left\|\theta_{t+1} - \tilde{\theta}_{PQ}\right\|^2 + |\lambda_{t+1} - \lambda^*|^2 \leq |\lambda_0 - \lambda^*|^2 + \left\|\theta_0 - \tilde{\theta}_{PQ}\right\|^2 + 2\eta^2\hat{G}_\lambda^2 + \eta^2(1 + \sqrt{2}\rho + \lambda^*)^2\hat{G}_\theta^2$$

$$+ 4\sqrt{2}\eta\rho\sqrt{T}\sigma_Q\sqrt{3\log\frac{2}{\tau}} + 2\sqrt{2}\eta\rho C_{PQ}(\log(t+2\kappa_Q)+2)^2 + T\gamma\eta(\lambda^*)^2. \tag{11}$$

Since we choose $\eta = \frac{c_\eta}{\sqrt{T}}$ and $\gamma = \hat{G}_\theta^2\eta$, where

$$c_\eta \leq \min\left\{\frac{\rho}{2\sqrt{2}\hat{G}_\lambda}, \frac{\rho}{2(1 + \sqrt{2}\rho + \lambda^*)\hat{G}_\theta}, \frac{\rho}{16\sqrt{6}\sigma_{PQ}\sqrt{\log\frac{2}{\tau}}}, \frac{\rho}{4C_{PQ}}\right\},$$

we conclude that $\left\|\theta_{t+1} - \tilde{\theta}_{PQ}\right\|^2 + |\lambda_{t+1} - \lambda^*|^2 \leq 2\rho^2$.

Now by similar analysis we have that for any $\lambda \geq 0$

$$\left\|\theta_{t+1} - \tilde{\theta}_{PQ}\right\|^2 + |\lambda_{t+1} - \lambda|^2 \leq \left\|\theta_t - \tilde{\theta}_{PQ}\right\|^2 + |\lambda_t - \lambda|^2 - 2\eta(L(\theta_t, \lambda) - L(\tilde{\theta}_{PQ}, \lambda_t))$$

$$+ 2\eta^2\hat{G}_\lambda^2 + \eta^2(1 + \lambda_t)^2\hat{G}_\theta^2$$

$$+ 2\eta\langle\ell(\theta_t; x_t, y_t) - \ell(\theta_{Q,t}; x_t, y_t) - (\ell(\theta_t) - \ell(\theta_{Q,t})), \lambda_t - \lambda\rangle$$

$$+ 2\eta\left\langle\ell(\hat{\theta}_Q) - \ell(\theta_{Q,t}), \lambda_t - \lambda\right\rangle + 2\eta r_t\left\|\theta_t - \tilde{\theta}_{PQ}\right\|$$

$$\leq \left\|\theta_t - \tilde{\theta}_{PQ}\right\|^2 + |\lambda_t - \lambda|^2 - 2\eta(L(\theta_t, \lambda) - L(\tilde{\theta}_{PQ}, \lambda_t))$$

$$+ 2\eta^2\hat{G}_\lambda^2 + \eta^2(1 + \lambda_t)^2\hat{G}_\theta^2$$

$$+ 2\eta r_t(\lambda_t + \lambda) + 2\eta h_t(\lambda_t + \lambda) + 2\sqrt{2}\eta\delta_t\rho.$$

Since $|\lambda_t - \lambda^*| \leq \sqrt{2}\rho$ we know $\lambda_t \leq \lambda^* + \sqrt{2}\rho$. Hence we have

$$\left\|\theta_{t+1} - \tilde{\theta}_{PQ}\right\|^2 + |\lambda_{t+1} - \lambda|^2 \leq |\lambda_t - \lambda|^2 + \left\|\theta_t - \tilde{\theta}_{PQ}\right\|^2 - 2\eta(L(\theta_t, \lambda) - L(\tilde{\theta}_{PQ}, \lambda_t))$$
$$+ 2\eta^2 \hat{G}_\lambda^2 + \eta^2(1 + \lambda_t)^2 \hat{G}_\theta^2 + 2\eta r_t \left(\lambda^* + \sqrt{2}\rho\right) + 2\eta h_t \left(\lambda^* + \sqrt{2}\rho\right)$$
$$+ 2\eta r_t \lambda + 2\eta h_t \lambda + 2\sqrt{2}\eta \delta_t \rho.$$

Performing telescoping sum yields:

$$\frac{1}{T} \sum_{t=0}^{T-1} L(\theta_t, \lambda) - L(\tilde{\theta}_{PQ}, \lambda_t) \leq \frac{1}{2\eta T}(|\lambda_0 - \lambda|^2 + \left\|\theta_0 - \tilde{\theta}_{PQ}\right\|^2) + \frac{1}{T}\eta \hat{G}_\lambda^2 + \frac{1}{2T}\eta \sum_{t=0}^{T-1}(1 + \lambda_t)^2 \hat{G}_\theta^2$$
$$+ \frac{1}{T} \sum_{t=0}^{T-1} r_t \left(\lambda^* + \sqrt{2}\rho\right) + \frac{1}{T} \sum_{t=0}^{T-1} h_t \left(\lambda^* + \sqrt{2}\rho\right) + \frac{1}{T} \sum_{t=0}^{T-1} r_t \lambda$$
$$+ \frac{1}{T} \sum_{t=0}^{T-1} h_t \lambda + \sqrt{2}\rho \frac{1}{T} \sum_{t=0}^{T-1} \delta_t.$$

By the definition of Lagrangian, we have

$$\frac{1}{T} \sum_{t=0}^{T-1}(\hat{R}_P(\theta_t) + \lambda g(\theta_t) - \frac{\gamma}{2}\lambda^2 - \hat{R}_P(\tilde{\theta}_{PQ}) - \lambda_t \underbrace{g(\tilde{\theta}_{PQ})}_{\leq 0} + \frac{\gamma}{2}\lambda_t^2)$$

$$\leq \frac{1}{2\eta T}(|\lambda_0 - \lambda|^2 + \left\|\theta_0 - \tilde{\theta}_{PQ}\right\|^2) + \frac{1}{T}\eta \hat{G}_\lambda^2 + \frac{1}{2T}\eta \sum_{t=0}^{T-1}(1 + \lambda_t)^2 \hat{G}_\theta^2$$

$$+ \frac{1}{T} \sum_{t=0}^{T-1} r_t \left(\lambda^* + \sqrt{2}\rho\right) + \frac{1}{T} \sum_{t=0}^{T-1} h_t \left(\lambda^* + \sqrt{2}\rho\right) + \frac{1}{T} \sum_{t=0}^{T-1} r_t \lambda + \frac{1}{T} \sum_{t=0}^{T-1} h_t \lambda + \sqrt{2}\rho \frac{1}{T} \sum_{t=0}^{T-1} \delta_t.$$

Evoking the bound from (9) and (10) yields:

$$\frac{1}{T} \sum_{t=0}^{T-1}(\hat{R}_P(\theta_t) + \lambda g(\theta_t) - \frac{\gamma}{2}\lambda^2 - \hat{R}_P(\tilde{\theta}_{PQ}) + \frac{\gamma}{2}\lambda_t^2)$$

$$\leq \frac{1}{2\eta T}(|\lambda_0 - \lambda|^2 + \left\|\theta_0 - \tilde{\theta}_{PQ}\right\|^2) + \frac{1}{T}\eta \hat{G}_\lambda^2 + \frac{1}{2T}\eta \sum_{t=0}^{T-1}(1 + \lambda_t)^2 \hat{G}_\theta^2$$

$$+ \frac{1}{T}\sqrt{T\hat{G}_\lambda^2}\sqrt{3\log\frac{2}{\tau}}\left(\lambda^* + 2\sqrt{2}\rho\right) + \frac{1}{T}C_{PQ} \cdot (\log(T + 2\kappa_Q) + 2)^2 \left(\lambda^* + \sqrt{2}\rho\right)$$

$$+ \frac{1}{T}\sqrt{T\hat{G}_\lambda^2}\sqrt{3\log\frac{2}{\tau}}\lambda + \frac{1}{T}C_{PQ} \cdot (\log(T + 2\kappa_Q) + 2)^2 \lambda + \frac{\sqrt{2}\rho\sigma_{PQ}\sqrt{3\log\frac{2}{\tau}}}{\sqrt{T}}.$$

Plugging in $\lambda_0 = 0$, $\theta_0 = \mathbf{0}$ and re-arranging the terms yields:

$$\frac{1}{T} \sum_{t=0}^{T-1}(\hat{R}_P(\theta_t) - \hat{R}_P(\tilde{\theta}_{PQ})) + \frac{1}{T} \sum_{t=0}^{T-1} \lambda g(\theta_t) - \left(\frac{\gamma}{2} + \frac{1}{2\eta T}\right)\lambda^2$$

$$\leq \frac{\rho^2}{\eta T} + \eta \hat{G}_\lambda^2 + \frac{1}{2T} \sum_{t=0}^{T-1}(\eta(1 + \lambda_t)^2 \hat{G}_\theta^2 - \gamma \lambda_t^2) + \frac{\sqrt{2}\rho\sigma_{PQ}\sqrt{3\log\frac{2}{\tau}}}{\sqrt{T}}$$

$$+ \frac{\hat{G}_\lambda\sqrt{3\log\frac{2}{\tau}}}{\sqrt{T}}(\lambda^* + 2\sqrt{2}\rho) + \frac{1}{T}C_{PQ}(\log(T + 2\kappa_Q) + 2)^2(\lambda^* + 2\sqrt{2}\rho)$$

$$+ \left(\frac{\hat{G}_\lambda\sqrt{3\log\frac{2}{\tau}}}{\sqrt{T}} + \frac{C_{PQ}(\log(T + 2\kappa_Q) + 2)^2}{T}\right)\lambda.$$

By our choice, $\gamma = \hat{G}_\theta^2 \eta$, so we have

$$\frac{1}{T}\sum_{t=0}^{T-1}\left(\hat{R}_P(\theta_t) - \hat{R}_P(\tilde{\theta}_{PQ})\right) + \frac{1}{T}\sum_{t=0}^{T-1}\lambda g(\theta_t) - \left(\frac{\gamma}{2} + \frac{1}{2\eta T}\right)\lambda^2$$

$$\leq \frac{\rho^2}{\eta T} + \eta\hat{G}_\lambda^2 + \eta\hat{G}_\theta^2 + \frac{\hat{G}_\lambda\sqrt{3\log\frac{2}{\tau}}}{\sqrt{T}}(\lambda^* + \sqrt{2}\rho) + \frac{\sqrt{2}\rho\sigma_{PQ}\sqrt{3\log\frac{2}{\tau}}}{\sqrt{T}}$$

$$+ \frac{1}{T}C_{PQ}(\log(T + 2\kappa_Q) + 2)^2(\lambda^* + \sqrt{2}\rho) + \left(\frac{\hat{G}_\lambda\sqrt{3\log\frac{2}{\tau}}}{\sqrt{T}} + \frac{C_{PQ}(\log(T + 2\kappa_Q) + 2)^2}{T}\right)\lambda.$$

Define $\hat{\theta}_T = \frac{1}{T}\sum_{t=0}^{T-1}\theta_T$, and then by Jensen's inequality we have

$$\left(\hat{R}_P(\bar{\theta}_T) - \hat{R}_P(\tilde{\theta}_{PQ})\right) + \lambda(\hat{R}_Q(\bar{\theta}_T) - \hat{R}_Q(\hat{\theta}_Q) - 6\epsilon_Q) - \left(\frac{\gamma}{2} + \frac{1}{2\eta T}\right)\lambda^2$$

$$\leq \frac{\rho^2}{\eta T} + \eta\hat{G}_\lambda^2 + \eta\hat{G}_\theta^2 + \frac{\hat{G}_\lambda\sqrt{3\log\frac{2}{\tau}}}{\sqrt{T}}(\lambda^* + \sqrt{2}\rho) + \frac{\sqrt{2}\rho\sigma_{PQ}\sqrt{3\log\frac{2}{\tau}}}{\sqrt{T}}$$

$$+ \frac{1}{T}C_{PQ}(\log(T + 2\kappa_Q) + 2)^2(\lambda^* + \sqrt{2}\rho) + \left(\frac{\hat{G}_\lambda\sqrt{3\log\frac{2}{\tau}}}{\sqrt{T}} + \frac{C_{PQ}(\log(T + 2\kappa_Q) + 2)^2}{T}\right)\lambda.$$

$\square$

## 13.2 Proof of Theorem 2

*Proof.* Note that Lemma 11 holds for any $\lambda \geq 0$. Now let's discuss by cases. If $\bar{\theta}_T$ is in the constraint set, then $\hat{\theta}_{PQ} = \bar{\theta}_T$ and we simply set $\lambda = 0$ and get the convergence:

$$\hat{R}_P(\hat{\theta}_{PQ}) - \hat{R}_P(\tilde{\theta}_{PQ})$$

$$\leq \frac{\rho^2}{\eta T} + \eta\hat{G}_\lambda^2 + \eta\hat{G}_\theta^2 + \frac{\hat{G}_\lambda\sqrt{3\log\frac{2}{\tau}}}{\sqrt{T}}\left(\lambda^* + 2\sqrt{2}\rho\right) + \frac{\sqrt{2}\rho\sigma_{PQ}\sqrt{3\log\frac{2}{\tau}}}{\sqrt{T}}$$

$$+ \frac{1}{T}C_{PQ}\cdot(\log(T + 2\kappa_Q) + 2)^2\left(\lambda^* + \sqrt{2}\rho\right).$$

If $\bar{\theta}_T$ is not in the constraint set, we set $\lambda = \frac{\hat{R}_Q(\bar{\theta}_T) - \hat{R}_Q(\hat{\theta}_Q) - 6\epsilon_Q}{\gamma + \frac{1}{\eta T}}$, and define $g(\theta) := \hat{R}_Q(\theta) - \hat{R}_Q(\hat{\theta}_Q) - 6\epsilon_Q$ for notational simplicity, which yields:

$$(\hat{R}_P(\bar{\theta}_T) - \hat{R}_P(\tilde{\theta}_{PQ})) + \frac{(g(\bar{\theta}_T))^2}{2(\gamma + \frac{1}{\eta T})}$$

$$\leq \frac{\rho^2}{\eta T} + \eta\hat{G}_\lambda^2 + \eta\hat{G}_\theta^2 + \frac{\hat{G}_\lambda\sqrt{3\log\frac{2}{\tau}}}{\sqrt{T}}\left(\lambda^* + 2\sqrt{2}\rho\right) + \frac{\sqrt{2}\rho\sigma_{PQ}\sqrt{3\log\frac{2}{\tau}}}{\sqrt{T}}$$

$$+ \frac{1}{T}C_{PQ}\cdot(\log(T + 2\kappa_Q) + 2)^2\left(\lambda^* + \sqrt{2}\rho\right)$$

$$+ \left(\frac{\hat{G}_\lambda\sqrt{3\log\frac{2}{\tau}}}{\sqrt{T}} + \frac{C_{PQ}\cdot(\log(T + 2\kappa_Q) + 2)^2}{T}\right)\left|\frac{g(\bar{\theta}_T)}{\gamma + \frac{1}{\eta T}}\right|. \qquad (12)$$

Since $\bar{\theta}_T$ is not in the constraint set and $\hat{\theta}_{PQ}$ is the projection of it onto inexact constraint set $\tilde{g}(\theta) := \hat{R}_Q(\theta) - \hat{R}_Q(\theta_{Q,T}) - 6\epsilon_Q + \epsilon_0 \leq 0$, by KKT condition we know $\tilde{g}(\hat{\theta}_{PQ}) = 0$ and $\bar{\theta}_T - \hat{\theta}_{PQ} = s \cdot \nabla\tilde{g}(\hat{\theta}_{PQ})$ for some $s > 0$. Defining $\Delta := \epsilon_0 - (\hat{R}_Q(\theta_{Q,T}) - \hat{R}_Q(\hat{\theta}_Q))$, and due to

our choice of $T$ we know $\Delta \geq 0$. Then we have

$$
\begin{aligned}
g(\bar{\theta}_T) &= g(\bar{\theta}_T) - \tilde{g}(\hat{\theta}_{PQ}) \\
&= \tilde{g}(\bar{\theta}_T) - \tilde{g}(\hat{\theta}_{PQ}) - (\tilde{g}(\bar{\theta}_T) - g(\bar{\theta}_T)) \\
&= \tilde{g}(\bar{\theta}_T) - \tilde{g}(\hat{\theta}_{PQ}) - (\hat{R}_Q(\hat{\theta}_Q) - \hat{R}_Q(\theta_{Q,T}) + \epsilon_0) \\
&\geq \left\langle \nabla \tilde{g}(\hat{\theta}_{PQ}), \bar{\theta}_T - \hat{\theta}_{PQ} \right\rangle - \Delta = \left\| \nabla \tilde{g}(\hat{\theta}_{PQ}) \right\| \left\| \bar{\theta}_T - \hat{\theta}_{PQ} \right\| - \Delta
\end{aligned}
$$

where the inequality is due to convexity of $g(\cdot)$. Evoking Lemma 8 with $\epsilon = 6\epsilon_Q - \epsilon_0 + \hat{R}_Q(\theta_{Q,T}) - \hat{R}_Q(\hat{\theta}_Q)$ gives that $\min_{\tilde{g}(\theta)=0} \left\| \nabla \tilde{g}(\hat{\theta}_{PQ}) \right\| \geq \sqrt{\lambda_{\min}^+(\hat{\Sigma}_Q)(6\epsilon_Q - \epsilon_0 + \hat{R}_Q(\theta_{Q,T}) - \hat{R}_Q(\hat{\theta}_Q))} \geq \sqrt{\lambda_{\min}^+(\hat{\Sigma}_Q)3\epsilon_Q}$, so $g(\bar{\theta}_T) \geq \sqrt{\lambda_{\min}^+(\hat{\Sigma}_Q)3\epsilon_Q} \left\| \bar{\theta}_T - \hat{\theta}_{PQ} \right\| - \Delta$, where the second inequality is due to that $\epsilon_0 \leq 3\epsilon_Q$.

On the other hand, since $\hat{\theta}_{PQ}$ is the projection of $\bar{\theta}_T$ onto constraint set, and $\tilde{\theta}_{PQ}$ is in the constraint set, we know

$$
\left\| \hat{\theta}_{PQ} - \tilde{\theta}_{PQ} \right\|^2 \leq \left\| \bar{\theta}_T - \tilde{\theta}_{PQ} \right\|^2 \leq 2\rho^2.
$$

Hence $\hat{\theta}_{PQ}$ also falls in the set $\left\{ \theta : \left\| \theta - \tilde{\theta}_{PQ} \right\|^2 \leq 2\rho^2 \right\}$, so the gradient upper bound $\hat{G}_\theta$ applies to $\hat{\theta}_{PQ}$. Hence we also know

$$
\begin{aligned}
g(\bar{\theta}_T) &= g(\bar{\theta}_T) - \tilde{g}(\hat{\theta}_{PQ}) \\
&= \hat{R}_Q(\bar{\theta}_T) - \hat{R}_Q(\hat{\theta}_Q) - 6\epsilon_Q - (\hat{R}_Q(\hat{\theta}_{PQ}) - \hat{R}_Q(\theta_{Q,T}) - 6\epsilon_Q + \epsilon_0) \\
&\leq \hat{G}_\theta \left\| \bar{\theta}_T - \hat{\theta}_{PQ} \right\| - \epsilon_0.
\end{aligned}
$$

Plugging the upper and lower bound of $g(\bar{\theta}_T)$ into (12) yields:

$$
\begin{aligned}
(\hat{R}_P(\bar{\theta}_T) &- \hat{R}_P(\tilde{\theta}_{PQ})) + \sqrt{T} \frac{\left( \sqrt{\lambda_{\min}^+(\hat{\Sigma}_Q)3\epsilon_Q} \left\| \bar{\theta}_T - \hat{\theta}_{PQ} \right\| - \Delta \right)^2}{4(c_\eta \hat{G}_\theta^2 + \frac{1}{c_\eta})} \\
&\leq \frac{\rho^2}{\eta T} + \eta \hat{G}_\lambda^2 + \eta \hat{G}_\theta^2 + \frac{\hat{G}_\lambda \sqrt{3 \log \frac{2}{\tau}}}{\sqrt{T}} \left( \lambda^* + 2\sqrt{2}\rho \right) + \frac{\sqrt{2}\rho \sigma_{PQ} \sqrt{3 \log \frac{2}{\tau}}}{\sqrt{T}} \\
&\quad + \frac{1}{T} C_{PQ} \cdot (\log(T + 2\kappa_Q) + 2)^2 \left( \lambda^* + \sqrt{2}\rho \right) \\
&\quad + \left( \frac{\hat{G}_\lambda \sqrt{3 \log \frac{2}{\tau}}}{\sqrt{T}} + \frac{C_{PQ} \cdot (\log(T + 2\kappa_Q) + 2)^2}{T} \right) \frac{\sqrt{T}}{c_\eta \hat{G}_\theta^2 + \frac{1}{c_\eta}} \hat{G}_\theta \left\| \bar{\theta}_T - \hat{\theta}_{PQ} \right\|.
\end{aligned}
$$

Notice the following decomposition:

$$
\hat{R}_P(\bar{\theta}_T) - \hat{R}_P(\tilde{\theta}_{PQ}) \geq \hat{R}_P(\bar{\theta}_T) - \hat{R}_P(\hat{\theta}_{PQ}) \geq -\hat{G}_\theta \left\| \bar{\theta}_T - \hat{\theta}_{PQ} \right\|.
$$

Also notice the fact $(a - b)^2 \geq \frac{1}{2}a^2 - b^2$ holding for any $a > 0, b > 0$, we know

$$
\left( \sqrt{\lambda_{\min}^+(\hat{\Sigma}_Q)3\epsilon_Q} \left\| \bar{\theta}_T - \hat{\theta}_{PQ} \right\| - \Delta \right)^2 \geq \frac{1}{2} \lambda_{\min}^+(\hat{\Sigma}_Q)3\epsilon_Q \left\| \bar{\theta}_T - \hat{\theta}_{PQ} \right\|^2 - \Delta^2.
$$

Putting pieces together yield the following inequality:

$$a \left\| \bar{\theta}_T - \hat{\theta}_{PQ} \right\|^2 - b \left\| \bar{\theta}_T - \hat{\theta}_{PQ} \right\| - c \leq 0,$$

where:

$$a = \frac{3\sqrt{T}\lambda_{\min}^+(\hat{\Sigma}_Q)\epsilon_Q}{8(c_\eta \hat{G}_\theta^2 + \frac{1}{c_\eta})}$$

$$b = \frac{\hat{G}_\theta}{(c_\eta \hat{G}_\theta^2 + \frac{1}{c_\eta})} \left( \hat{G}_\lambda \sqrt{3\log \frac{2}{\tau}} + \frac{C_{PQ} \cdot (\log(T + 2\kappa_Q) + 2)^2}{\sqrt{T}} \right) + \hat{G}_\theta,$$

$$c = \frac{\frac{\rho^2}{c_\eta} + c_\eta(\hat{G}_\lambda^2 + \hat{G}_\theta^2) + \hat{G}_\lambda \sqrt{3\log \frac{2}{\tau}} \left(\lambda^* + 2\sqrt{2}\rho\right) + \sqrt{2}\rho\sigma_{PQ}\sqrt{3\log \frac{2}{\tau}}}{\sqrt{T}}$$

$$+ \frac{C \cdot (\log(T + 2\kappa_Q) + 2)^2 \left(\lambda^* + \sqrt{2}\rho\right)}{T} + \sqrt{T}\frac{\Delta^2}{4(c_\eta \hat{G}_\theta^2 + \frac{1}{c_\eta})}.$$

Assume $T \geq \left( \frac{C_{PQ} \cdot (\log(T+2\kappa_Q)+2)^2}{\hat{G}_\lambda \sqrt{3\log \frac{2}{\tau}}} \right)^2$, so $b \leq \frac{2\hat{G}_\theta \hat{G}_\lambda \sqrt{3\log \frac{2}{\tau}}}{(c_\eta \hat{G}_\theta^2 + \frac{1}{c_\eta})} + \hat{G}_\theta$. Solving the above quadratic inequality yields:

$$\left\| \bar{\theta}_T - \hat{\theta}_{PQ} \right\| \leq \frac{b + \sqrt{b^2 + 4ac}}{2a} \leq \frac{b}{a} + \sqrt{\frac{c}{a}}$$

$$\leq \frac{8(c_\eta \hat{G}_\theta^2 + \frac{1}{c_\eta})\hat{G}_\theta + 16\hat{G}_\theta \hat{G}_\lambda \sqrt{3\log \frac{2}{\tau}}}{3\lambda_{\min}^+(\hat{\Sigma}_Q)\epsilon_Q \sqrt{T}} + \frac{\Delta}{\sqrt{\lambda_{\min}^+(\hat{\Sigma}_Q)\epsilon_Q}}$$

$$+ \sqrt{\frac{8(c_\eta \hat{G}_\theta^2 + \frac{1}{c_\eta})}{3\sqrt{T}\lambda_{\min}^+(\hat{\Sigma}_Q)\epsilon_Q}} \sqrt{\frac{\frac{\rho^2}{c_\eta} + c_\eta(\hat{G}_\lambda^2 + \hat{G}_\theta^2) + \hat{G}_\lambda \sqrt{3\log \frac{2}{\tau}} \left(\lambda^* + 2\sqrt{2}\rho\right) + \sqrt{2}\rho\sigma_{PQ}\sqrt{3\log \frac{2}{\tau}}}{\sqrt{T}}}$$

$$+ \sqrt{\frac{8(c_\eta \hat{G}_\theta^2 + \frac{1}{c_\eta})}{3\sqrt{T}\lambda_{\min}^+(\hat{\Sigma}_Q)\epsilon_Q}} \sqrt{\frac{C_{PQ} \cdot (\log(T + 2\kappa_Q) + 2)^2 \left(\lambda^* + \sqrt{2}\rho\right)}{T}}$$

$$= \frac{16(c_\eta \hat{G}_\theta^2 + \frac{1}{c_\eta})\hat{G}_\theta + 16\hat{G}_\theta \hat{G}_\lambda \sqrt{3\log \frac{2}{\tau}}}{\lambda_{\min}^+(\hat{\Sigma}_Q)\epsilon_Q \sqrt{T}} + \frac{\Delta}{\sqrt{\lambda_{\min}^+(\hat{\Sigma}_Q)\epsilon_Q}} + \frac{C_{PQ} \cdot (\log(T + 2\kappa_Q) + 2)^2 \left(\lambda^* + \sqrt{2}\rho\right)}{2T}$$

$$+ \frac{\frac{\rho^2}{c_\eta} + c_\eta(\hat{G}_\lambda^2 + \hat{G}_\theta^2) + \hat{G}_\lambda \sqrt{3\log \frac{2}{\tau}} \left(\lambda^* + 2\sqrt{2}\rho\right) + \sqrt{2}\rho\sigma_{PQ}\sqrt{3\log \frac{2}{\tau}}}{2\sqrt{T}}$$

where at the last step we use the fact $\sqrt{ab} \leq \frac{a^2 + b^2}{2}$. Finally, note the following decomposition:

$$\hat{R}_P(\hat{\theta}_{PQ}) - \hat{R}_P(\tilde{\theta}_{PQ}) = \hat{R}_P(\hat{\theta}_{PQ}) - \hat{R}_P(\bar{\theta}_T) + \hat{R}_P(\bar{\theta}_T) - \hat{R}_P(\tilde{\theta}_{PQ})$$

$$\leq \hat{G}_\theta \left\| \bar{\theta}_T - \hat{\theta}_{PQ} \right\| + \hat{R}_P(\bar{\theta}_T) - \hat{R}_P(\tilde{\theta}_{PQ})$$

$$\leq \left( \hat{G}_\theta + \frac{2\hat{G}_\lambda \sqrt{3\log \frac{2}{\tau}}}{c_\eta \hat{G}_\theta^2 + \frac{1}{c_\eta}} \hat{G}_\theta \right) \left\| \bar{\theta}_T - \hat{\theta}_{PQ} \right\|$$

$$+ \frac{\frac{\rho^2}{c_\eta} + c_\eta(\hat{G}_\lambda^2 + \hat{G}_\theta^2) + \hat{G}_\lambda \sqrt{3\log \frac{2}{\tau}} \left(\lambda^* + 2\sqrt{2}\rho\right) + \sqrt{2}\rho\sigma_{PQ}\sqrt{3\log \frac{2}{\tau}}}{\sqrt{T}}$$

$$+ \frac{C_{PQ} \cdot (\log(T + 2\kappa_Q) + 2)^2 \left(\lambda^* + \sqrt{2}\rho\right)}{T}.$$

Since we assume $T \geq T \geq \left( \frac{4C_{PQ} \cdot (\log(T+2\kappa_Q))^2}{\hat{G}_\lambda \sqrt{\lambda_{\min}^+(\hat{\Sigma}_Q)\epsilon_Q \log 1/\tau}} \right)^2$, we know

$$\hat{R}_P(\hat{\theta}_{PQ}) - \hat{R}_P(\tilde{\theta}_{PQ}) \leq \hat{G}_\theta \left\| \bar{\theta}_T - \hat{\theta}_{PQ} \right\| + \hat{R}_P(\bar{\theta}_T) - \hat{R}_P(\tilde{\theta}_{PQ})$$

$$\leq \left( \hat{G}_\theta + \frac{2\hat{G}_\lambda \sqrt{3\log\frac{2}{\tau}}}{c_\eta \hat{G}_\theta^2 + \frac{1}{c_\eta}} \hat{G}_\theta \right) \left\| \bar{\theta}_T - \hat{\theta}_{PQ} \right\|$$

$$+ \frac{\frac{\rho^2}{c_\eta} + c_\eta(\hat{G}_\lambda^2 + \hat{G}_\theta^2) + 2\hat{G}_\lambda \sqrt{3\log\frac{2}{\tau}} \left( \lambda^* + 2\sqrt{2}\rho \right) + \sqrt{2}\rho\sigma_{PQ}\sqrt{3\log\frac{2}{\tau}}}{\sqrt{T}}.$$

Plugging bound of $\left\| \bar{\theta}_T - \hat{\theta}_{PQ} \right\|$ and Lemma 9 yields:

$$\hat{R}_P(\hat{\theta}_{PQ}) - \hat{R}_P(\tilde{\theta}_{PQ})$$

$$\leq \left( \hat{G}_\theta + \frac{2\hat{G}_\lambda \sqrt{3\log\frac{2}{\tau}}}{c_\eta \hat{G}_\theta^2 + \frac{1}{c_\eta}} \hat{G}_\theta \right) \left( \frac{16(c_\eta \hat{G}_\theta^2 + \frac{1}{c_\eta})\hat{G}_\theta + 16\hat{G}_\theta \hat{G}_\lambda \sqrt{3\log\frac{2}{\tau}}}{\lambda_{\min}^+(\hat{\Sigma}_Q)\epsilon_Q \sqrt{T}} + \frac{2C_{PQ}\log T}{(T+2\kappa_Q)} + \frac{2\epsilon_0}{\sqrt{\lambda_{\min}^+(\hat{\Sigma}_Q)\epsilon_Q}} \right)$$

$$+ 48 \left( \hat{G}_\theta + \frac{\hat{G}_\lambda \sqrt{\log\frac{2}{\tau}}}{c_\eta \hat{G}_\theta^2 + \frac{1}{c_\eta}} \hat{G}_\theta \right) \left( \frac{\frac{\rho^2}{c_\eta} + c_\eta(\hat{G}_\lambda^2 + \hat{G}_\theta^2) + \hat{G}_\lambda \sqrt{\log\frac{2}{\tau}} (\lambda^* + \rho) + \rho\sigma_{PQ}\sqrt{\log\frac{2}{\tau}}}{\sqrt{T}} \right).$$

Since we choose $\epsilon_0 = \frac{C_{PQ}}{T+2\kappa_Q}$, we know $\frac{2\epsilon_0}{\sqrt{\lambda_{\min}^+(\hat{\Sigma}_Q)\epsilon_Q}} = \frac{2C_{PQ}}{\sqrt{\lambda_{\min}^+(\hat{\Sigma}_Q)\epsilon_Q}(T+2\kappa_Q)}$.

Now we simplify the above bound. By the definition of $c_\eta$ we know $c_\eta \leq \frac{1}{\hat{G}_\theta}$, so we have

$$\hat{R}_P(\hat{\theta}_{PQ}) - \hat{R}_P(\tilde{\theta}_{PQ})$$

$$\leq \left( \hat{G}_\theta + \frac{2\hat{G}_\lambda \sqrt{3\log\frac{2}{\tau}}}{c_\eta \hat{G}_\theta + 1} \right) \left( \frac{16(\hat{G}_\theta + \frac{1}{c_\eta})\hat{G}_\theta + 16\hat{G}_\theta \hat{G}_\lambda \sqrt{3\log\frac{2}{\tau}}}{\lambda_{\min}^+(\hat{\Sigma}_Q)\epsilon_Q \sqrt{T}} + \frac{4C_{PQ}\log T}{\sqrt{\lambda_{\min}^+(\hat{\Sigma}_Q)\epsilon_Q}(T+2\kappa_Q)} \right)$$

$$+ 48 \left( \hat{G}_\theta + \frac{\hat{G}_\lambda \sqrt{\log\frac{2}{\tau}}}{c_\eta \hat{G}_\theta + 1} \right) \left( \frac{\frac{\rho^2}{c_\eta} + c_\eta(\hat{G}_\lambda^2 + \hat{G}_\theta^2) + \hat{G}_\lambda \sqrt{\log\frac{2}{\tau}} (\lambda^* + \rho) + \rho\sigma_{PQ}\sqrt{\log\frac{2}{\tau}}}{\sqrt{T}} \right)$$

$$\leq \left( \hat{G}_\theta + 2\hat{G}_\lambda \sqrt{3\log\frac{2}{\tau}} \right) \left( \frac{16(\hat{G}_\theta + \frac{1}{c_\eta})\hat{G}_\theta + 16\hat{G}_\theta \hat{G}_\lambda \sqrt{3\log\frac{2}{\tau}}}{\lambda_{\min}^+(\hat{\Sigma}_Q)\epsilon_Q \sqrt{T}} + \frac{4C_{PQ}\log T}{(T+2\kappa_Q)} \right)$$

$$+ 48 \left( \hat{G}_\theta + \hat{G}_\lambda \sqrt{\log\frac{2}{\tau}} \right) \left( \frac{\frac{\rho^2}{c_\eta} + c_\eta(\hat{G}_\lambda^2 + \hat{G}_\theta^2) + \left( (\lambda^* + \rho)\hat{G}_\lambda + \rho\sigma_{PQ} \right) \sqrt{\log\frac{2}{\tau}}}{\sqrt{T}} \right).$$

Again recall we choose: $T \geq \left( \frac{4C_{PQ} \cdot (\log(T+2\kappa_Q))^2}{\hat{G}_\lambda \sqrt{\lambda_{\min}^+(\hat{\Sigma}_Q)\epsilon_Q \log 1/\tau}} \right)^2$, so $\frac{4C_{PQ}\log T}{\sqrt{\lambda_{\min}^+(\hat{\Sigma}_Q)\epsilon_Q}(T+2\kappa_Q)} \leq \frac{\hat{G}_\lambda \sqrt{\log 1/\tau}}{\sqrt{T}}$.
So we have

$$\hat{R}_P(\hat{\theta}_{PQ}) - \hat{R}_P(\tilde{\theta}_{PQ})$$

$$\leq \left( \hat{G}_\theta + 2\hat{G}_\lambda \sqrt{3 \log \frac{2}{\tau}} \right) \left( \frac{32\frac{\hat{G}_\theta}{c_\eta} + 16\hat{G}_\theta \hat{G}_\lambda \sqrt{3 \log \frac{2}{\tau}}}{\lambda_{\min}^+(\hat{\Sigma}_Q)\epsilon_Q \sqrt{T}} + \frac{\hat{G}_\lambda \sqrt{\log \frac{1}{\tau}}}{\sqrt{T}} \right)$$

$$+ 2 \left( \hat{G}_\theta + 2\hat{G}_\lambda \sqrt{3 \log \frac{2}{\tau}} \right) \left( \frac{\frac{\rho^2}{c_\eta} + c_\eta(G_\lambda^2 + \hat{G}_\theta^2) + \sqrt{2}\rho\sigma_{PQ} \sqrt{3 \log \frac{2}{\tau}}}{\sqrt{T}} \right)$$

$$\lesssim \left( \hat{G}_\theta + \hat{G}_\lambda \sqrt{\log \frac{1}{\tau}} \right) \left( \frac{\frac{\hat{G}_\theta}{c_\eta} + \hat{G}_\theta \hat{G}_\lambda \sqrt{\log \frac{1}{\tau}}}{\lambda_{\min}^+(\hat{\Sigma}_Q)\epsilon_Q \sqrt{T}} + \frac{((\lambda^* + \rho)\hat{G}_\lambda + \rho\sigma_{PQ}) \sqrt{\log \frac{1}{\tau}} + \frac{\rho^2}{c_\eta} + c_\eta \hat{G}_\lambda^2}{\sqrt{T}} \right).$$

Finally, by definition of $c_\eta$ we know $\frac{\rho^2}{c_\eta} \geq c_\eta \hat{G}_\lambda^2$, $\frac{\rho^2}{c_\eta} \geq \rho\hat{G}_\lambda$ and $\frac{\rho^2}{c_\eta} \geq \rho 16\sqrt{6}\sigma_{PQ} \sqrt{\log \frac{2}{\tau}}$ which concludes the proof:

$$\hat{R}_P(\hat{\theta}_{PQ}) - \hat{R}_P(\tilde{\theta}_{PQ}) \lesssim \left( \hat{G}_\theta + \hat{G}_\lambda \sqrt{\log \frac{1}{\tau}} \right) \left( \frac{\frac{\hat{G}_\theta}{c_\eta} + \hat{G}_\theta \hat{G}_\lambda \sqrt{\log \frac{1}{\tau}}}{\lambda_{\min}^+(\hat{\Sigma}_Q)\epsilon_Q \sqrt{T}} + \frac{\lambda^* \hat{G}_\lambda \sqrt{\log \frac{1}{\tau}} + \frac{\rho^2}{c_\eta}}{\sqrt{T}} \right).$$

$\square$

## 14 Proof of the Results of General Loss

In this section we provide the missing proofs in Section 9. We first introduce the following lemma which establishes the convergence of auxiliary iterates $\theta_{Q,t}$ to $Q$ ERM model.

**Lemma 12** (High probability convergence of $\theta_{Q,t}$). *If we choose $\alpha_t = \frac{1}{m_1} \cdot \frac{1}{t+2\kappa}$, then with probability at least $1 - \tau$, for any $t \geq 0$ we have:*

$$\hat{R}_Q(\theta_{Q,t}) - \hat{R}_Q(\hat{\theta}_Q) \leq \frac{m_1(1 + 2\kappa_Q)(\frac{2}{m_1^2}\|\nabla \hat{R}_Q(\theta_{Q,0})\|^2 + 2\|\theta_{Q,0}\|^2)}{t + 2\kappa} + \frac{6\hat{G}_\theta^2 \log(2/\tau)(\log t + 1)}{m_1(t + 2\kappa)}.$$

*Proof.* We first examine the boundedness of $\|\theta_{Q,t}\|$. Define $e_t \doteq \nabla \ell(\theta_{Q,t}; x_t, y_t) - \nabla \ell(\theta_{Q,0}; x_t, y_t)$. According to updating rule of $\theta_{Q,t}$ we have

$$\begin{aligned}
\|\theta_{Q,t+1} - \theta_{Q,0}\| &\leq \|\theta_{Q,t} - \alpha_t \nabla \ell(\theta_{Q,t}; x_t, y_t) - (\theta_{Q,0} - \alpha_t \nabla \ell(\theta_{Q,0}; x_t, y_t))\| \\
&\quad + \alpha_t \|\nabla \ell(\theta_{Q,0}; x_t, y_t)\| \\
&= \sqrt{\|\theta_{Q,t} - \theta_{Q,0}\|^2 - 2\alpha_t \langle e_t, \theta_{Q,t} - \theta_{Q,0}\rangle + \alpha_t^2 \|e_t\|^2} \\
&\quad + \alpha_t \|\nabla \ell(\theta_{Q,0}; x_t, y_t)\| \\
&\leq \|\theta_{Q,t} - \theta_{Q,0}\| + \alpha_t \|\nabla \ell(\theta_{Q,0}; x_t, y_t)\| \\
&\leq \sum_{j=0}^{t} \alpha_j^2 \|\nabla \ell(\theta_{Q,0}; x_j, y_j)\|^2 \\
&\leq \sum_{j=0}^{t} \frac{1}{m_1} \cdot \frac{1}{t + 2\kappa} \sup_{(x,y) \in S_Q} \|\nabla \ell(\theta_{Q,0}; x, y)\| \\
&\leq \frac{1 + \log(T + 2\kappa)}{m_1} \sup_{(x,y) \in S_Q} \|\nabla \ell(\theta_{Q,0}; x, y)\| \quad (13)
\end{aligned}$$

where the third step is due to the co-coercivity of the gradient of the convex and smooth functions:

$$\langle \nabla \ell(\theta_{Q,t}; x_t, y_t) - \nabla \ell(\theta_{Q,0}; x_t, y_t), \theta_{Q,t} - \theta_{Q,0}\rangle \geq \frac{1}{m_2} \|\nabla \ell(\theta_{Q,t}; x_t, y_t) - \nabla \ell(\theta_{Q,0}; x_t, y_t)\|^2.$$

Hence we can bound $\|\theta_{Q,t}\|$ as

$$\|\theta_{Q,t}\| \le \|\theta_{Q,t} - \theta_{Q,0}\| + \|\theta_{Q,0}\|$$
$$\le \frac{1 + \log(T + 2\kappa)}{m_1} \sup_{(x,y) \in S_Q} \|\nabla\ell(\theta_{Q,0}; x, y)\|.$$

Hence we can compute sub-Gaussian parameter. By the definition of $\hat{G}_\theta$

$$\left\|\nabla\ell(\theta_{Q,t}; x_t, y_t) - \nabla\hat{R}_Q(\theta_{Q,t})\right\| \le 2 \sup_{(x,y) \in S_Q} \|\nabla\ell(\theta_{Q,t}; x, y)\| \le 2\hat{G}_\theta$$

According to Proposition 2, we know $\left\|\nabla\ell(\theta_{Q,t}; x_t, y_t) - \nabla\hat{R}_Q(\theta_{Q,t})\right\|$ is $\hat{G}_\theta^2$ sub-Gaussian.

Now, we evoke the result from Theorem 3.7 from [29] that if the gradient noise is $\hat{G}_\theta^2$ sub-Gaussian, then with our choice of $\alpha_t$, with probability at least $1 - \tau$ it holds for any integer $t > 0$ that

$$\hat{R}_Q(\theta_{Q,t}) - \hat{R}_Q(\hat{\theta}_Q) \le \frac{m_1(1 + 2\kappa_Q)\|\hat{\theta}_Q\|^2}{t + 2\kappa} + \frac{6\hat{G}_\theta^2 \log(2/\tau)(\log t + 1)}{m_1(t + 2\kappa)}.$$

We further bound $\|\hat{\theta}_Q\|^2$ as

$$\|\hat{\theta}_Q\|^2 \le 2\|\hat{\theta}_Q - \theta_{Q,0}\|^2 + 2\|\theta_{Q,0}\|^2$$
$$\le \frac{4}{m_1}(\hat{R}_Q(\theta_{Q,0}) - \hat{R}_Q(\hat{\theta}_Q)) + 2\|\theta_{Q,0}\|^2$$
$$\le \frac{4}{m_1}(\hat{R}_Q(\theta_{Q,0}) - \min_{\theta \in \mathbb{R}^D} \hat{R}_Q(\theta)) + 2\|\theta_{Q,0}\|^2$$
$$\le \frac{4}{m_1} \cdot \frac{1}{2m_1} \left\|\nabla\hat{R}_Q(\theta_{Q,0})\right\|^2 + 2\|\theta_{Q,0}\|^2$$

which concludes the proof.

$\square$

### 14.1 Proof of Theorem 4

*Proof.* The proof is almost identical to that of Theorem 2. In Section 13.2, choosing $C_{PQ} = m_1(1 + 2\kappa)(\frac{2}{m_1^2}\|\nabla\hat{R}_Q(\theta_{Q,0})\|^2 + 2\|\theta_{Q,0}\|^2) + \frac{6\hat{G}_\theta^2 \log(2/\tau)}{m_1}$ and plugging in $\tilde{g}(\theta) = \hat{R}_Q(\theta) - \hat{R}_Q(\theta_{Q,T}) - 3\epsilon_Q + \epsilon_0$, then we have

$$g(\bar{\theta}_T) = g(\bar{\theta}_T) - \tilde{g}(\hat{\theta}_{PQ})$$
$$= \tilde{g}(\bar{\theta}_T) - \tilde{g}(\hat{\theta}_{PQ}) - (\tilde{g}(\bar{\theta}_T) - g(\bar{\theta}_T))$$
$$= \tilde{g}(\bar{\theta}_T) - \tilde{g}(\hat{\theta}_{PQ}) - (\hat{R}_Q(\hat{\theta}_Q) - \hat{R}_Q(\theta_{Q,T}) + \epsilon_0)$$
$$\ge \left\langle \nabla\tilde{g}(\hat{\theta}_{PQ}), \bar{\theta}_T - \hat{\theta}_{PQ} \right\rangle - \Delta = \left\|\nabla\tilde{g}(\hat{\theta}_{PQ})\right\| \left\|\bar{\theta}_T - \hat{\theta}_{PQ}\right\| - \Delta$$

Let $\theta_0$ be such that $g(\theta_0) = 0$, and then we have

$$\min_{\tilde{g}(\theta)=0} \|\nabla g(\theta)\| \ge \min_{\tilde{g}(\theta)=0} \|\nabla g(\theta_0)\| - \|\nabla g(\theta_0) - \nabla g(\theta)\|$$
$$\ge r(3\epsilon_Q) - 2m_1 (g(\theta_0) - g(\theta)) = r - 2m_1 g(\theta_0)|_{\tilde{g}(\theta)=0}$$
$$= r(3\epsilon_Q) - 2m_1(\epsilon_0 - (\hat{R}_Q(\theta_{Q,T}) - \hat{R}_Q(\hat{\theta}_Q)))$$
$$\ge r(3\epsilon_Q) - 2m_1\epsilon_0$$

, so $g(\bar{\theta}_T) \ge (r(3\epsilon_Q) - 2m_1\epsilon_0) \left\|\bar{\theta}_T - \hat{\theta}\right\| - \Delta$. Since we choose $\epsilon_0 = \frac{C_{PQ}}{T}$ and $T \ge \frac{4C_{PQ}m_1}{r(3\epsilon_Q)}$, we know $r(3\epsilon_Q) - 2m_1\epsilon_0 \ge \frac{1}{2}r(3\epsilon_Q)$. The rest of the proof follows the same way.

$\square$

## 14.2 Proof of Proposition 1

*Proof.* By the standard Rademacher complexity analysis (see [26]) we know:

$$\sup_{h \in \mathcal{H}} |R_\mu(\theta) - \hat{R}_\mu(\theta)| \leq 2\mathcal{R}_n(\ell \circ \mathcal{H}) + M_\ell \sqrt{\frac{\log \frac{2}{\tau}}{2n_\mu}}.$$

Plugging in the upper bound of $\mathcal{R}_n(\ell \circ \mathcal{H})$ from Assumption 5 concludes the proof. $\quad\square$

## 14.3 Proof of Corollary 1

*Proof.* First, since $\hat{\theta}_{PQ} \in \{\theta : \hat{R}_Q(\theta) - \hat{R}_Q(\hat{\theta}_Q) \leq 3\epsilon_Q\}$, then by Proposition 1 and our choice of $\epsilon_Q$ we know $\mathcal{E}_Q(\hat{\theta}_{PQ}) \leq 5\epsilon_Q$ with probability at least $1 - \tau$ over the randomness of $S_Q$.

Then we discuss by cases. If $\theta_P^* \in \{\theta : \hat{R}_Q(\theta) - \hat{R}_Q(\hat{\theta}_Q) \leq 3\epsilon_Q\}$, then since $\hat{R}_P(\hat{\theta}_{PQ}) - \hat{R}_P(\tilde{\theta}_{PQ}) \leq \epsilon_P$, by Proposition 1 and our choice of $\epsilon_P$ we know with probability at least $1 - 2\tau$ over the randomness of $S_P$ and Algorithm 3,

$$\begin{aligned}
\mathcal{E}_P(\hat{\theta}_{PQ}) &= R_P(\hat{\theta}_{PQ}) - R_P(\theta_P^*) \\
&= R_P(\hat{\theta}_{PQ}) - \hat{R}_P(\hat{\theta}_{PQ}) + \hat{R}_P(\hat{\theta}_{PQ}) - \hat{R}_P(\tilde{\theta}_{PQ}) + \underbrace{\hat{R}_P(\tilde{\theta}_{PQ}) - \hat{R}_P(\theta_P^*)}_{\leq 0} \\
&\quad + \hat{R}_P(\theta_P^*) - R_P(\theta_P^*) \leq 3\epsilon_P.
\end{aligned}$$

Hence $\mathcal{E}_Q(\hat{\theta}_{PQ}) \leq \delta(3\epsilon_P)$.

If $\theta_P^* \notin \{\theta : \hat{R}_Q(\theta) - \hat{R}_Q(\hat{\theta}_Q) \leq 3\epsilon_Q\}$, then we know $\mathcal{E}_Q(\theta_P^*) \geq \epsilon_Q$ with probability at least $1 - \tau$ over the randomness of $S_Q$. This is because for any $\theta$ such that $\mathcal{E}_Q(\theta) \leq \epsilon_Q$, it must be in the set $\{\theta : \hat{R}_Q(\theta) - \hat{R}_Q(\hat{\theta}_Q) \leq 3\epsilon_Q\}$. To see this, note that

$$\hat{\mathcal{E}}_Q(\theta) \leq \mathcal{E}_Q(\theta) + 2\epsilon_Q \leq 3\epsilon_Q.$$

Hence we know

$$\mathcal{E}_Q(\hat{\theta}_{PQ}) \leq 5\epsilon_Q \leq 5\mathcal{E}_Q(\theta_P^*) \leq 5\delta(\epsilon_P).$$

Putting piece together we have with probability at least $1 - 3\tau$, it holds that

$$\mathcal{E}_Q(\hat{\theta}_{PQ}) \leq 5 \min \{\epsilon_Q, \delta(3\epsilon_P)\}.$$

$\quad\square$

