# OpenReview forum: "Mixed-Sample SGD: an End-to-end Analysis of Supervised Transfer Learning"
_NeurIPS.cc/2025/Conference — NeurIPS 2025 poster_

### Official Review · Reviewer_cNej · 2025-07-02

**Clarity:** 3
**Significance:** 2
**Originality:** 2
**Rating:** 3
**Confidence:** 2

**Summary:**

This paper provides a theoretical analysis of the supervised transfer learning settings between the source and target distributions of a linear regression model. This paper shows that the returned solution is extrapolated between the source and target datasets, leaning towards the beneficial one, under the mixed-sample SGD setting. Experiments confirm the theoretical analysis

**Questions:**

Please see the weaknesses

**Ethical Concerns:**

["NO or VERY MINOR ethics concerns only"]

**Final Justification:**

This paper gives a theoretical understanding of the optimisation procedure to balance the contribution of source and target during training. Despite several interesting results, the original setting of this work departs from the practical setup, which limits the contribution of this work.

**Limitations:**

Please see the weaknesses.

**Paper Formatting Concerns:**

No formatting concerns

**Quality:**

2

**Strengths And Weaknesses:**

This paper has several strengths:
- A new theoretical result on the transfer learning setting with a linear regression (classification) model.
- Experiments confirm theoretical results.


Weaknesses:
- This paper considers a simple setting where the source and target datasets share the same domain $\mathcal{X}$ and $\mathcal{Y}$. This setting is too strict and less applicable in practice since the transfer learning is often done between domains or datasets with the same input domain but have different labels.

- The "usefulness" of the source dataset is indicated by the sample size, which is not always the case. In my opinion, it is chosen since this paper only focuses on the source and target datasets from the same domain for both input and output.

- The presentation of this paper causes confusion, and it is not ready to appear at this venue. I recommend that authors carefully revise the manuscript to make it more readable.  For instance, Section 4.1. provides a series of lemmas and starting with an algorithm for warming up, the motivation behind the warm-up algorithm should be discussed, and the interpretation of lemmas should be also discussed carefully to understand why it is important. In addition, the importance of $\rho$, the norm between the initial and the calculated point, should be discussed.

- What is the motivation behind the use of mixed SGD?

- The related works should be discussed further why this work is important in the literature instead of listing lines of works.

---

> ### Author Rebuttal · Authors · 2025-07-31
>
> The reviewer has a question about the setting, which might be due to a lack of familiarity with the literature of theoretical transfer learning.
>
> We argue that assuming input and label spaces are shared across source and target distributions is very **standard** in the theoretical transfer learning papers. See seminal works [Mansour et al 2009] [Ben-David et al 2010] [Zhang et al 2019]. However, we agree with the reviewer that modern ML also has the scenarios where source and target share different label spaces, but that is a completely different setup to this paper. In those settings, people consider different model architectures, e.g., source and target share the same representation layers but with different output heads. This is beyond the scope of this paper.
>
> [Mansour et al 2009] Mansour, Yishay, Mehryar Mohri, and Afshin Rostamizadeh. "Domain adaptation: Learning bounds and algorithms." arXiv preprint arXiv:0902.3430 (2009).
>
>
> [Ben-David et al 2010] Ben-David, Shai, et al. "A theory of learning from different domains." Machine learning 79.1 (2010): 151-175.
>
> [Zhang et al 2019] Zhang, Yuchen, et al. "Bridging theory and algorithm for domain adaptation." International conference on machine learning. PMLR, 2019.
>
> **Q1: The "usefulness" of the source dataset is indicated by the sample size, which is not always the case.**
>
> We respectfully disagree with your opinion that the usefulness of source data is measured solely by the sample size. The quality of source data is also measured by how close the two distributions are, indicated by $\lambda_{\max} (\Sigma_P^{-1} \Sigma_Q)$ and $\mathcal{E}_Q(\theta^*_P)$, which are also reflected by our main Theorem.
>
> **Q2: The interpretations of the Lemmas in Section 4.1.**
>
> Since the stepsize choice depends on a bunch of quantities: upper bound of initial distance to optimal model ($\rho$), gradient upper bound ($\hat G_\theta$ and $\hat G_\lambda$), and upper bound ($\lambda^\star$) (please see the conditions in Theorem 2), it is necessary to provide approaches to estimate them from the data, which are given by Lemma 1 to 4. In Lemma 1 and 4, we can see that the upper bound of $\rho$ and $\lambda^\star$ depends on $\hat \theta_Q$, so we use a warm-up algorithm to approximate it. We will make the purpose of that section clearer.
>
>
> **Q3: The motivation behind the use of mixed SGD?**
>
> The proposal of the algorithm is motivated by the following fact: a desired *adaptive* optimization algorithm for transfer learning should automatically detect whether the source data or the target data are more useful and sample the data from them accordingly. The probability of sampling from source or target should also be adjusted adaptively, based on the performance of current model on the target data. Our algorithm, inspired by the Lagrangian objective, exactly works in this way.
>
> **Q4: The related works should be discussed further why this work is important in the literature instead of listing lines of works.**
>
> We discussed the importance of our work compared to prior work multiple times in the Introduction. In a nutshell, the previous works in transfer learning mainly focus on the statistical aspect of this problem, while none of them gives a concrete optimization procedure, that can achieve optimal statistical guarantee. Our work fills this blank.

---

> ### Author Response · Authors · 2025-08-06
>
> Thanks for reviewing our paper. Did our rebuttal address your concerns? Please let us know if you have any other questions. We are happy to answer them before the discussion period ends.

---

> ### Comment · Reviewer_cNej · 2025-08-07
>
> I thank the authors for addressing my concerns; however, my main concern remains in the impractical setting of this work. Another comment to improve the manuscript is to add descriptions for the notations in the Figure 1 caption, which is at the very top of the paper, so we do not have to go back and forth to understand them. I raised my score accordingly.

---

> > ### Author Response · Authors · 2025-08-09
> >
> > We appreciate your suggestion and reconsideration. We will incorporate your suggestions in the revised version.

---

### Official Review · Reviewer_7uqy · 2025-07-03

**Clarity:** 3
**Significance:** 3
**Originality:** 3
**Rating:** 5
**Confidence:** 2

**Summary:**

The paper proposes mixed-sample sgd which adaptively alternates between sampling from the source and target datasets at a rate that automatically adjusts based on the source data's quality. This approach avoids computationally expensive tuning the source data's importance weight while tracking a sequence of constrained convex programs to maintain strong theoretical guarantees. The authors prove that their method converges at a rate of $1/\sqrt{T}$ to a solution that automatically performs as well as a model trained on either the source or target data alone, whichever is better.

**Questions:**

Why is $\hat{\theta}_Q$ unknown? Since we're performing linear regression with finite samples, I would expect a closed-form solution to be available.

**Ethical Concerns:**

["NO or VERY MINOR ethics concerns only"]

**Final Justification:**

During the rebuttal, the authors addressed other reviewers' concerns. I don't have any remaining concerns either. So I keep my score as 5.

**Limitations:**

please see the weaknesses.

**Quality:**

3

**Strengths And Weaknesses:**

Disclaimer: I am not expert in optimization so most of my reviews are educated guess. If there are something wrong, please correct me.

## Strengths
- The paper provides rigorous proof for the proposed algorithm's convergence at a standard $O(1/\sqrt{T})$ rate and, more importantly, its adaptivity.

- The paper is logically structured and does an excellent job in the introduction of motivating the problem, highlighting a clear gap in the existing literature.

- The paper tackles the core, practical challenge of safely and efficiently leveraging source data in transfer learning.

- The proposed Mixed-Sample SGD algorithm is original. Its mechanism of using a dynamic, learnable sampling rate to approximate a constrained optimization problem is a new approach compared to prior methods that rely on expensive cross-validation or direct projections.

## Weaknesses




- The formal guarantees are derived for convex loss functions. This is a significant limitation, as it doesn't cover the non-convex nature of modern deep learning models. The authors acknowledge this, but the gap between the convex theory and some of the deep-learning-based experiments remains.

- The detailed proofs contain minor  but recurring errors in the calculation of constants, particularly regarding factors of 2 and 4 when bounding gradient norms. Since empirical risk is  $\frac{1}{n} \sum_{ (x_i, y_i) \in S_Q} (\theta^\top x_i - y_i)^2$,
the gradient should be
$$\nabla\hat{R}_Q(\theta) = 2\hat{\Sigma}_Q(\theta-\hat{\theta}_Q)$$

in lemma 8. Thus, the minimum gradient norm should be $4\epsilon \lambda^{+}_{\min}(\hat{\Sigma}_Q)$. Similarly, the factor of 2 is omitted in line 582.

- A minor typo: unversal -> universal in line 148.

---

> ### Author Rebuttal · Authors · 2025-07-31
>
> Thank you for your time and positive comments. We will try to address your questions as follows.
>
> **Q1: The results are restricted to convex loss**
>
> For nonconvex risk functions, it might be intractable in terms of convergence, but the statistical guarantee for the solution of the proposed QCQP will still hold.
>
> **Q2: Why not using a closed form solution of $\hat\theta_Q$**
>
>  First, we wanted to present a procedure that applies beyond linear regression where a closed form solution may not exist.
>
> In the case of linear regression, you are right that a closed form solution may be plugged in, but our goal is *to account for all computation*: in large dimension settings, the closed form solution requires iterative power methods to approximate which may cost $O(d^3)$ in the worst case; instead we approximate the closed-form solution in parallel and account for all computation in our results. As such, we never need to compute it exactly, especially as the algorithm may end up preferring to rely on the source when it's most informative.
> We also apologize for the typos and will fix them. Thanks for catching them.

---

> ### Comment · Reviewer_7uqy · 2025-08-05
>
> - Could you elaborate on how does the statistical guarantee for the solution of QCQP still hold?
>
> - Regarding, the constant factor 2 or 4, could you confirm that it is a mistake?

---

> ### Author Response · Authors · 2025-08-06
>
> Thanks for your post-rebuttal response. We try to address your concern as follows.
>
>
> **Q1. Could you elaborate on how does the statistical guarantee for the solution of QCQP still hold?**
>
> We suppose that the reviewer is referring to the limit of the QCQP sequence in Equation (2). If so, as stated in Section 3, Page 4, the statistical guarantees of our procedures are in fact obtained by relating it to the guarantees of this limiting QCQP (of the form $\min \hat R_P$ subject to $\hat R_Q \lesssim \hat R_Q^\star + \epsilon_Q$).
>
> For intuition, consider the simplest case where $\theta^\star_P = \theta^\star_Q$, and suppose $\tilde \theta_{PQ}$ is a solution to the above limiting QCQP; then the constraint ensures that $\tilde \theta_{PQ}$ has good performance under $Q$, while the minimization of $\hat R_P$ ensures that it also has good performance under $P$, hence the guarantees of $\min\\{\tilde \epsilon_P, \epsilon_Q\\}$, i.e., interpolating between $\tilde \epsilon_P$ and $\epsilon_Q$; here $\tilde \epsilon_P$ stands for the performance under $Q$ of predictors with small $P$ error $\epsilon_P$.
>
> **Q2. Regarding, the constant factor 2 or 4, could you confirm that it is a mistake?**
>
> Yes in Lemma 8 the factor in the gradient term should be 2. We will fix them. Thanks for catching this.

---

> > ### Comment · Reviewer_7uqy · 2025-08-09
> >
> > Thank you for the clarification. I keep my score as it is.

---

### Official Review · Reviewer_2HaD · 2025-07-03

**Clarity:** 2
**Significance:** 2
**Originality:** 2
**Rating:** 4
**Confidence:** 3

**Summary:**

The paper provided an analysis of the expected error when using mixed-sample SGD algorithm in the context of supervise transfer learning. The main result is presented in Theorem 1, that provided an upper bound of the expected loss at the target distribution. More particular, it includes $\varepsilon_Q$ (best achievable rate from $Q$ target distribution) and some quantities measure the discrepancy between source and target distributions. Other theorems (2,3) and Lemmas (1,2,3,4,5,6,7) were stated to support the proof of Theorem 1.
In the work, the author presented an hybrid algorithm that use both source and target data to train the model, meanwhile, the ratio between their sizes was  adjusted along the training procedure. The main reason for mixing data in the SGD is that the source-target problem has two minimization procedures at the same time, that is connected through a parameter in the Lagrangian form. The authors proposed to use mixed data with adaptive rate to deal with the situation.

**Questions:**

Why is there $6\epsilon_Q$ in the constraint (2), I mean the factor $6$?

Does $\lambda_{\max}  (\Sigma_p^{-1} \Sigma_Q)$ exist?

Notation $\eta = \frac{c_\eta}{\sqrt{T}}$ in Theorem 1, so $\eta$ depends on $c_\eta$?

How to check all conditions do not violate each other?

Should have a table for all quantities to check?

$\theta_{Q,t}$ what does $t$ means here, step $t$?  equation (2)

Line 46, what does $\tilde{\nabla}$ mean?

Line 59, what does ``converge in risk" mean?

Line 142, what is $\theta_0$?

In Theorem 2, why do we need condition $M_x \geq 1$? Because we could be in the situation when $\|x\| \leq 1$.

Theorem 1 includes $\epsilon_Q, \epsilon_P$ but no  definition for those terms. When I track back line 72, some words describe them, still no mathematical definition.

What is the limit of $\lambda_t$ as $t\rightarrow \infty$?

**Ethical Concerns:**

["NO or VERY MINOR ethics concerns only"]

**Final Justification:**

I keep my score unchanged  due to the poor presentation and limited experiments.

**Limitations:**

Yes

I would like to reject the paper since the poor presentation of its mathematics and the results seem to be not technically novel and challenging to prove. However, I am not familiar with the previous theoretical works in this domain. Hence, I reserve the decision when more things will be clear.

**Quality:**

2

**Strengths And Weaknesses:**

Strengths:

1. I think the work is concrete, when results are stated, algorithm was derived, the motivation is well-stated.
2. The flow of the proofs is clear, in which the order and link between steps were explained.

Weaknesses:

1. The linear regression setting is too simple. Two distributions are differed only by their covariance matrices.
2. Poor presentation of mathematical results. It seems to me that the author assumed that readers know the problem in advance. I have a list of questions below.
3. I am not certain that the proof is novel. The main idea is common technique in analysis and probability, consider a quantities in its local neighborhood and for others outside the neighborhood bounding it by using other properties.
4. I am not certain that the upper  bound is Theorem 1 is optimal? The RHS of the inequality in Theorem 1 has two terms: one with respect to target distribution, other measures the discrepancy between source and target distribution. This is not surprising, since the estimator comes from mixed data.
5. Experiments were carried out on small datasets, synthetic data and school data with very small sample sizes.  It seems to me that the authors consider it is a theoretical work. Meanwhile, the more important thing is if the proposed algorithm actually works/ outperforms other methods in various settings and deployed to help solving problems.

---

> ### Author Rebuttal · Authors · 2025-07-31
>
> The reviewer has questions about technical novelty, but this is perhaps due to a lack of familiarity with analyses on the subject matter.
>
> We re-emphasize that the main novelty of the work is in deriving a provably efficient SGD procedure that manages to recover optimal statistical rates of transfer, which is very much not trivial. The main difficulties are outlined in between Line 60 and 71 in the Introduction, which the reviewer perhaps missed. We are happy to expand on this further if the reviewer has more specific questions.
>
> **Q1: The linear regression setting is too simple and two distributions differ only by their covariance matrices.**
>
> The reviewer seems to have missed many important points:
>
> --- As stated repeatedly, our results apply beyond linear regression, to general convex settings (see Theorem 4 in Appendix 8), and we chose to focus on linear regression in the main body for clarity of exposition.
>
> --- In the linear regression case, **we do not restrict to changes in covariance**: we very specifically allow for changes in regression functions, as displayed in our statistical results and explained on page 5. What's surprising indeed is that the procedure can adapt to all unknown such changes, including changes in regression functions.
>
> **Q2: Optimality of Theorem 1**
>
> Yes Theorem 1 is the optimal rate for transfer learning in linear regression setting. The matching lower bound is given in the recent arXiv paper [Hanneke and Kpotufe 2024, Theorem 3].
>
> **Q3: More Experiments on the large-scale dataset**
>
>
> We agree with you that the paper would benefit from evaluations on larger datasets.
> We have additional results on a slightly larger dataset (see experiment on Berkeley Yearbook Dataset, in Appendix 9.1, where $n_P$ ranges from $500$ to $1300$). We will add more results on the Malware-iot dataset (a large-scale malicious network traffic detection dataset) in the revised version.
>
>
>  We however would like to emphasize that the focus of this work is primarily theoretical (i.e., show the feasibility of an SGD with optimal STL guarantees), and our focus in these first experiments was to show the practical promise of these insights.
>
>
>
> **Q4: Technical questions**
>
> 4.1 Why is there a factor of $6$ in Eq. 2?
>
> We choose the factor $6$ for proof convenience. We require our constraint set to have two properties (Lemma 5) :(1)  any $\theta$ in the constraint set will have a small Q risk, and (ii) any $\theta$ that has a
>  small Q risk is covered by our constraint set with high probability. To ensure these properties
> we require the factor to be $6$. Please check the proof of Lemma 5 for more details.
>
> 4.2 Does $\lambda_{\max}(\Sigma_P^{-1}\Sigma_Q)$ exist?
>
> Yes $\lambda_{\max}(\Sigma_P^{-1}\Sigma_Q)$ always exists. The reason is that  $ \Sigma_P^{-1}\Sigma_Q$ and $ \Sigma_P^{-1/2}\Sigma_Q\Sigma_P^{-1/2}$ share the same eigenvalues, and $\Sigma_P^{-1/2}\Sigma_Q\Sigma_P^{-1/2}$ is a symmetric matrix so all these eigenvalues are real. Hence $\lambda_{\max}(\Sigma_P^{-1}\Sigma_Q)$ always exists.
>
> 4.3 Does $\eta$ depend on $c_\eta$?
>
> Yes $\eta = \frac{c_\eta}{\sqrt{T}}$ and $c_\eta$ is a factor that hides the dependence of the learning rate on the parameters $\hat G_\theta, \hat G_\lambda, \lambda^*$ and $\rho$.
>
>
> 4.4 How to check all conditions do not violate each other?
>
> We are not sure which conditions you refer to. If you refer to the two conditions for $\hat\sigma$ and $c_\eta$ in Theorem 2, they can hold simultaneously, since the condition of $\hat\sigma$ does not depend on $c_\eta$.
>
> 4.5 What is the meaning of $t$ in $\theta_{Q,t}$ in Eq. 2?
>
> $t$ denotes $t$th iteration. $\theta_{Q,t}$ are the iterates we maintained to approximate $\hat\theta_{Q}$ during algorithm proceeding, since we do not have $\hat{\theta}_{Q}$ in hand.
>
> 4.6 What is $\tilde{\nabla}$?
>
> It denotes the stochastic gradient operator.
>
> 4.7 What does 'converge in risk' mean?
>
> It means $\hat R_P(\hat\theta_{PQ}) \mapsto \hat R_P(\tilde\theta_{PQ})$ as the total number of iterations goes to infinity.
>
> 4.8 What is $\theta_0$?
>
> It is the initialization of the algorithm.
>
> 4.9 In Theorem 2 why do we need $M_x \geq 1$?
>
> We assume this only for simplifying the bound. It can be removed without affecting our main result.
>
> 4.10 No definitions for $\epsilon_P$ and $\epsilon_Q$.
>
> We defined these quantities. Please see Line 148.
>
> 4.11 What is the limit of $\lambda_t$?
>
> $\lambda_t$ may not have a limit, since in convex-concave minimax problem, the last iterate could diverge under gradient descent and ascent dynamics, but the average of the iterates, $\frac{1}{T}\sum_{t=1}^T \lambda_t$ converges to the saddle point $\lambda^\star$, when $T$ goes to infinity.

---

> > ### Comment · Reviewer_2HaD · 2025-08-05
> > **Reply to the rebuttal**
> >
> > I would like to thank the authors for their clarification.
> >
> > My experience for the mathematical presentation of this work is bad. For example, $\epsilon_Q$ appears  in equation (2), but only defined inside Theorem 1, one page later. Similarly, if I want to check the definitions of $\epsilon_P$ and $\epsilon_Q$ in Theorem 1,  then where I could find $\sigma_y^2$. $\eta = \frac{c_\eta}{\sqrt{T}}$, if $\eta$ is defined by $c_\eta$, then the constant $c_{\eta}$ should not have $\eta$. Again, in Theorem 3, we need to understand that $\epsilon_P$ and $\epsilon_Q$ come from Theorem 1.  No explanation for the appearance of Definition 3, lacking discussion/comments for Lemmas, lacking explained structures for proofs presented in the Appendix etc.
> >
> > The author emphasized on the importance of the algorithm, but experiments limited on the small data set, even the data sets mentioned in the Appendix are still small.
> >
> > I keep my score unchanged, leaning to accepted, since I believe that the mathematical work is correct. I hope that the bar for   mathematical presentation in theoretical paper should be raised.

---

> ### Author Response · Authors · 2025-08-06
>
> Thanks for your post-rebuttal response. We try to address your concern as follows.
>
>
> **Q1. Regarding definitions**
>
> We actually explained on page 1 already that $\epsilon_Q, \epsilon_P$ will stand for the best rate achievable using $Q$ data alone before it appears in equation (2). In that equation and subsequent discussions the exact setting of $\epsilon_Q$ is irrelevant as it may be viewed simply as a parameter to the procedure. In Theorem 1, we derive the exact setting that leads to the guarantees.
>
> While we believe this is standard practice in theoretical papers, we also understand that the reader may not immediately see that we are using $\epsilon_Q, \epsilon_P$ as place holders for rates to be explicitly derived later. We will add an explicit notation section where the roles of these variables are emphasized early.
>
> We tried our best to discuss lemmas and proof structure within the main body, given the page limits; however, given the reviewer's questions we understand that the discussion should be extended, which fortunately is possible given the extended page limit for the camera ready version.

---

### Official Review · Reviewer_vDwt · 2025-07-03

**Clarity:** 2
**Significance:** 3
**Originality:** 4
**Rating:** 5
**Confidence:** 4

**Summary:**

This paper studies a class of supervised transfer learning algorithms that utilize stochastic gradient descent (SGD) to construct a regressor by minimizing the empirical risk on a source domain while constraining the empirical risk on a target domain. The authors provide a theoretical analysis of the generalization error for this approach, showing that the SGD solution converges to that of a specific hypothesis transfer learning algorithm with a well-established generalization rate. Notably, the resulting generalization error adapts between the rates achieved by training solely on the source or solely on the target data. The authors support their theoretical findings with experiments, demonstrating that their proposed algorithm achieves adaptive performance, matching the rates of either solo source or solo target training depending on the scenario.

**Questions:**

1. Could the authors please provide a more detailed comparison with Hanneke et al. (2024)? Specifically, I would appreciate clarification on how your theoretical framework differs from theirs, and what novel contributions your paper makes beyond their work. A clearer articulation of these distinctions would significantly strengthen my evaluation of the paper's originality and significance.

2. Regarding the experimental evaluation, could the authors provide more details about the implementation of all compared methods? In particular, I noticed that the HTL method shows constant accuracy regardless of source sample size, which seems counterintuitive. Additionally, the runtime comparisons would be more convincing with implementation details to ensure fair comparison (e.g., number of iterations, optimization parameters). Addressing these concerns would improve my assessment of the paper's experimental quality.

**Ethical Concerns:**

["NO or VERY MINOR ethics concerns only"]

**Final Justification:**

I lean toward accepting this paper primarily because it proposes a practical algorithm for transfer learning with rigorous theoretical error guarantees. It offers significant practical benefits, particularly by reducing computational cost through the use of a first-order optimization method with adaptivity.

**Limitations:**

yes

**Paper Formatting Concerns:**

I have no concerns about the formatting of the paper.

**Quality:**

3

**Strengths And Weaknesses:**

I lean toward accepting this paper primarily due to the benefit of proposing a practical algorithm for transfer learning with rigorous theoretical error guarantees.

This paper is clearly written and addresses the relevant topic of error analysis for the SGD method in transfer learning. The theoretical and experimental contributions are well-highlighted, making the main results easily understandable. The theoretical results are supported by rigorous mathematical proofs that appear sound upon review. The clarity of the paper is generally high, with the exception of comparisons with existing work.

The paper provides error upper bounds for the mixed-sample SGD algorithm in a transfer learning setup. While the obtained error bounds match those shown by Hanneke et al. (2024), the originality of this paper lies in analyzing these bounds for the more practical mixed-sample SGD algorithm. The significance of this paper hinges on the extent of advancement beyond Hanneke et al. (2024), but unfortunately, the authors do not provide detailed comparisons with this prior work. They merely mention that Hanneke et al. (2024) focuses on classification settings, when in fact that work provides a general framework applicable to both classification and regression. Indeed, the theoretical framework of Hanneke et al. (2024) has wider applicability than this paper, and regarding error bounds, the results are equivalent. This lack of detailed comparison diminishes the clarity regarding the paper's significance.

The primary contribution may lie in providing a concrete implementation to find a predictor within the confidence set defined by Hanneke et al. (2024), effectively offering a tractable implementation of Algorithm 1 from that work. Developing such a tractable algorithm represents a significant contribution to transfer learning. However, as mentioned, the paper lacks rigorous comparisons with Hanneke et al. (2024), including clarification of the relationship between the algorithms. A more thorough comparison would substantially improve the paper.

I have concerns about the experimental quality. First, the accuracies of the HTL method remain constant regardless of source sample size. The authors attribute this to HTL's difficulty in finding suitable regularization parameters through cross-validation due to limited target data. However, the relatively small error bars suggest limited impact from small sample sizes, contradicting this explanation. Without implementation details or code for HTL, I cannot verify the experimental design's adequacy. Second, runtime results are highly implementation-dependent, yet the authors provide no implementation details or code for HTL, PSGD, or their proposed methods. This raises concerns about potential unfair comparisons, such as the proposed algorithm potentially using fewer iterations than comparison algorithms. Providing implementation details would help establish the fairness of the runtime comparisons.

---

> ### Author Rebuttal · Authors · 2025-07-31
>
> Thank you for your positive and insightful comments. We notice that your main concerns are the comparison with prior work, and the implementation details of the experiments. We will try to address them as follows.
>
> **Q1: Regarding performance and implementation details of HTL**
>
> In fact, in some experiments, the performance of HTL does improve as $n_p$ increases, see, e.g., Figure 6 in the Appendix. We agree that our explanation for HTL's performance is poorly worded. Indeed, when source and target are too different, HTL is biased towards target largely so it gives similar performance as target ERM, while our method still automatically gains from the source data, and more importantly, our procedure is parameter-free---it does not require searching for a bias-parameter. For the implementation of HTL, we first use SGD to find the approximation of $\hat\theta_P$ as the reference model, then we perform a 5-fold cross-validation to select the best bias. Then we run SGD for $1000$ epochs to optimize the objective $\hat R_Q(\theta)+\lambda ||\theta -\hat\theta_P ||^2$. The same epoch number is also used for our algorithm and PSGD.
>
> **Q2: A more detailed comparison with [Hanneke and Kpotufe 2024]**
>
> While we achieve the same rate as in [Hanneke and Kpotufe 2024] (when specialized to regression), the constrained ERM procedure being approximated by SGD is different from the procedure proposed in [Hanneke and Kpotufe 2024] in the following respect: their procedure returns any element of the intersection of two confidence sets (so boils down to constraint satisfaction) while we consider a constrained ERM which from which it is easier to derive an SGD procedure. Our rates are of the same order, although our constants are worse. We agree with the reviewer that we should further emphasize this comparison in the work.
>
> We will make the above two points clearer, and we again thank you for your insightful comments.

---

> > ### Comment · Reviewer_vDwt · 2025-08-05
> >
> > Thank you for the authors’ responses.
> >
> > Could the authors clarify the rationale behind the statement: "when source and target are too different, HTL is biased towards target largely so it gives similar performance as target ERM"? From my perspective, it is not immediately clear why a well-tuned HTL would exhibit such a bias.

---

> ### Author Response · Authors · 2025-08-06
>
> Thanks for your post-rebuttal response. We try to address your concern as follows.
>
> **Q1. Could the authors clarify the rationale behind the statement: "when source and target are too different, HTL is biased towards target largely so it gives similar performance as target ERM"?**
>
>
> We would like to remind the reviewer that HTL aims to minimize an objective of the form $\hat R_Q(\theta) + \lambda ||\theta - \hat \theta_P||^2$ where $\theta_P$ is the model learned on the source. If the $\theta_P$ is too far from the optimal $\theta_Q^\star$, then HTL would prefer a small $\lambda$, i.e., bias towards the target. In particular, it does not fully use both datasets and only uses the source via the learned $\hat \theta_P$, while MS-SGD truly aggregates both samples and importantly always biases towards the source if there is not enough evidence that the source could be hurtful (as MS-SGD can be related to minimizing $\hat R_P$ subject to target penalization, i.e., $\hat R_Q$ within sampling variability). Suppose, for instance, that there are two good source models $\theta_{P, 1}$ and $\theta_{P, 2}$ that are source ERM  candidates, but $\theta_{P, 2}$ being bad under target. Then if $\hat \theta_P \approx \theta_{P, 2}$, $\hat \theta_P$ is far from $\theta^\star_Q$; the target penalization implicit in MS-SGD would instead quickly reject any solution close to $\theta_{P, 2}$ and keep minimizing $\hat R_Q$ to find better solution (here close to $\theta_{P, 1}$) if one exists.
>
> Finally we re-emphasize that one of our goals is to achieve good time complexity along with low target error. This is to say that HTL becomes more expansive as we refine the range of $\lambda$'s to tune over, which is a trade-off to keep in mind. We have no such tuning parameter, although $\epsilon_Q$ in our penalization needs to be chosen carefully, and the analysis suggests a setting of $O(d/n_Q)$ which works well in our experiments.

---

### Decision · Program_Chairs · 2025-09-17

**Decision:**

Accept (poster)

**Comment:**

This paper analyzes the mixed-sample SGD scenario for supervised transfer learning. The setting considers alternating between source and target data during sampling, without requiring prior knowledge of source data quality. The authors propose an adaptive sampling algorithm that maintains statistical transfer guarantees, allowing the method to benefit from informative sources while avoiding negative transfer.

The reviewers generally appreciate the significance of the main contribution: the proposed algorithm is backed by rigorous theoretical analysis while remaining practical enough for real-world transfer learning scenarios. While there were shared concerns regarding the practicality of the experimental setup, I believe the theoretical contribution outweighs these limitations.

Therefore, I recommend acceptance.